IPM/P-2023/59

# Pseudo Rényi Entanglement Entropies For an Excited State and Its Time Evolution in a 2D CFT

Farzad Omidi

*School of Physics, Institute for Research in Fundamental Sciences (IPM)*
*P.O. Box 19395-5531, Tehran, Iran*
and
*Kavli Institute for Theoretical Sciences (KITS),*
*University of Chinese Academy of Sciences, 100190 Beijing, P. R. China*

E-mail: farzad@ipm.ir

**Abstract**

In this paper, we investigate the second and third pseudo Rényi entanglement entropies (PREE) for a locally excited state $|\psi\rangle$ and its time evolution $|\phi\rangle = e^{-iHt}|\psi\rangle$ in a two-dimensional conformal field theory whose field content is a free massless scalar field. We consider excited states which are constructed by applying primary operators at time $t = 0$, on the vacuum state. We study the time evolution of the PREE for an entangling region in the shape of finite and semi-infinite intervals at zero temperature. It is observed that the PREE is always a complex number for $t \neq 0$ and is a pure real number at $t = 0$. Moreover, we discuss its dependence on the location $x_m$ of the center of the entangling region.

# 1 Introduction

In the past two decades, after the discovery of the celebrated Ryu-Takayanagi (RT) [1] and Hubeny-Rangamani-Takayanagi (HRT) formulae [2] in the context of the AdS/CFT correspondence [3–5], we have witnessed a deep connection between gravity and quantum information. These breakthroughs have shed light on the fascinating idea of how to construct the bulk geometry from the quantum entanglement of the corresponding state in the dual quantum field theory [6–8].

One of the important measures of quantum entanglement which has been investigated intensively, is the n-th Rényi entanglement entropy (REE). Consider a bipartite system with the density matrix $\rho$ and the Hilbert space $\mathcal{H} = \mathcal{H}_A \otimes \mathcal{H}_B$. By tracing out the states in $\mathcal{H}_B$, one can define a reduced

density matrix $\rho_A$ for the subsystem $A$ as follows

$$\rho_A = \mathrm{Tr}_B\left(\rho\right) = \sum_{i \in \mathcal{H}_B} \langle i|\rho|i\rangle. \tag{1.1}$$

Next, the n-th REE for the subsystem $A$ is defined by

$$S_A^{(n)} = \frac{1}{1-n} \log \mathrm{Tr}\left[(\rho_A)^n\right]. \tag{1.2}$$

It is well known that in the limit $n \to 1$, it reduces to entanglement entropy (EE). Moreover, this quantity is real valued and UV divergent. The Rényi entanglement entropies (REE's) of locally excited states constructed by applying primary [9–14] and descendant operators [15] or a combination of both of them [16] on the vacuum state in conformal field theories (CFTs), irrational CFTs [17], warped CFTs [18] and $T\overline{T}/J\overline{T}$ deformed CFTs [19] have been explored extensively in the literature.

Recently, an interesting generalization of REE dubbed "pseudo Rényi entanglement entropy" (PREE) was introduced in ref. [20]. Consider a bipartite system and choose two non-orthogonal pure states [1] $|\psi\rangle$ and $|\phi\rangle$ in the Hilbert space. One can define a transition matrix as follows

$$\tau^{\psi|\phi} = \frac{|\psi\rangle\langle\phi|}{\langle\phi|\psi\rangle}, \tag{1.3}$$

where it is normalized such that $\mathrm{Tr}[\tau^{\psi|\phi}] = 1$. By taking the trace of the transition matrix on the Hilbert space $\mathcal{H}_B$, one can define a reduced transition matrix

$$\tau_A^{\psi|\phi} = \mathrm{Tr}_B[\tau^{\psi|\phi}]. \tag{1.4}$$

Next, the n-th PREE is defined by the n-th REE for $\tau_A^{\psi|\phi}$ as follows [20]

$$S_A^{(n)} = \frac{1}{1-n} \log \mathrm{Tr}\left[(\tau_A^{\psi|\phi})^n\right]. \tag{1.5}$$

Since, the transition matrix is generally not Hermitian [2], PREE is usually complex valued. It is in contrast to REE which is a real quantity. Moreover, it can be applied to distinguish different quantum phases [22, 24, 25] and has several interesting properties [20]:

1. $S_A^{(n)}\left(\tau_A^{\psi|\phi}\right) = S_B^{(n)}\left(\tau_B^{\psi|\phi}\right)$.

2. when $B$ is empty, one has $S_A^{(n)}\left(\tau_A^{\psi|\phi}\right) = 0$.

3. if $|\psi\rangle$ has no entanglement, one obtains $S_A^{(n)}\left(\tau_A^{\psi|\phi}\right) = 0$.

4. for $|\phi\rangle = |\psi\rangle$, it reduces to the REE of the subsystem $A$.

Furthermore, by taking the limit $n \to 1$ in eq. (1.5), one obtains the pseudo entanglement entropy (PEE) of the subsystem $A$ [3]

$$S_A = -\mathrm{Tr}\left[\tau_A^{\psi|\phi} \log \tau_A^{\psi|\phi}\right]. \tag{1.6}$$

---

[1] Recently, the generalization of PREE to mixed states was investigated in ref. [21].

[2] More precisely, one has $\left(\tau^{\psi|\phi}\right)^\dagger = \tau^{\phi|\psi}$ [20].

[3] It was recently conjectured that time-like EE in CFTs as well as complex-valued EE in CFTs dual to de Sitter spacetimes can be regarded as PEE [31, 32, 34].

It should be pointed out that PEE and PREE have been studied extensively both in quantum field theory [20–31, 33–36] and holography [20, 24, 27, 30, 31, 34]. On the quantum field theory side, it can be calculated by the replica trick [37, 38]. On the other hand, on the gravity side, it can be computed by the area of a minimal surface in Euclidean time dependent backgrounds [20].

The aim of this paper is to study [4] the second and third PREE for an excited state $|\psi\rangle$ and its time evolution $|\phi\rangle = e^{-iHt}|\psi\rangle$ in a two-dimensional CFT (See eq. (3.1)). To be more precise, we calculate the difference

$$\Delta S_A^{(n)} = S^{(n)}(\tau_A^{\psi|\phi}) - S^{(n)}(\rho_A^{(0)}), \tag{1.7}$$

between the n-th PREE of the states $|\psi\rangle$ and $|\phi\rangle$, and the n-th REE $S^{(n)}(\rho_A^{(0)})$ of the vacuum state. Here, $\rho_A^{(0)} = tr_B(\rho^{(0)})$ is the reduced density matrix for the vacuum state $|0\rangle$.

The organization of the paper is as follows: in Section 2 we review the calculation of the second REE for locally excited states constructed by applying two primary operators in a two-dimensional CFT. In Section 3, we calculate the second PREE for two different excited states $|\psi\rangle$ and $|\phi\rangle$ in a two-dimensional CFT. In Section 4, we calculate the third PREE for the aforementioned states in the same CFT. In Section 5, we summarize our results and discuss future directions.

## 2 Second Rényi Entanglement Entropy for an Excited State in a $CFT_2$

In this section, we review the calculation of the second REE for some excited states constructed by applying two primary operators $\mathcal{O}_{1,2}$ on the vacuum state. This section is based on ref. [10] (See also refs. [9–11]). We consider a two-dimensional conformal field theory on a plane $\mathbb{R}^2$ whose field content is a free massless scalar field $\phi(x,t)$. We discuss the entangling regions in the shape of finite and semi-infinite intervals, respectively.

### 2.1 Finite Interval

We first study the case where the entangling region is a finite interval $A \in [0, L]$ at $t = 0$. We consider an excited state [5]

$$|\psi\rangle = e^{-iHt}e^{-aH}\mathcal{O}_i(x = -l)|0\rangle, \tag{2.1}$$

where $a$ is a UV cutoff since states with higher energies are suppressed more. Moreover, $\mathcal{O}_i(x,t)$ is a primary operator inserted at $x = -l < 0$ and $t = 0$, and $|0\rangle$ is the vacuum state. In the following, we just consider positive times, i.e. $t > 0$, after which the operator is inserted. To calculate, we analytically continue to Euclidean time $\tau = -it$ and define the complex coordinate $w = x + i\tau$. The density matrix is simply given by

$$\rho(t) = |\psi\rangle\langle\psi|$$

---

[4] While we were completing this work, ref. [39] was appeared on arXiv which studied a similar problem in the context of quantum mechanics and many body systems.

[5] We insert the operators $\mathcal{O}_i(t,x)$ at the time $t = 0$. For convenience, we do not write the time coordinate of the operators in what follows.

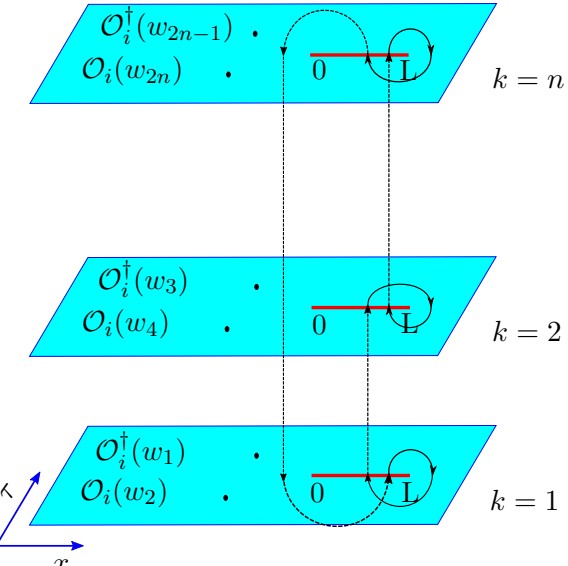

Figure 1: Replica trick for the calculation of the n-th REE of a locally excited state $|\psi\rangle$ in eq. (2.1). The entangling region is a finite interval $A \in [0, \infty]$. The difference with the replica trick for the n-th REE of the vacuum state, is that there are two operators $\mathcal{O}_i(w_{2k})$ and $\mathcal{O}_i^\dagger(w_{2k-1})$ on the k-th sheet.

$$= e^{-iHt} e^{-aH} \mathcal{O}_i(x = -l)|0\rangle\langle 0|\mathcal{O}_i^\dagger(x = -l) e^{-aH} e^{iHt}$$

$$= e^{H\tau} \mathcal{O}_i(x = -l)|0\rangle\langle 0|\mathcal{O}_i^\dagger(x = -l) e^{-H\tau}$$

$$= \mathcal{O}_i(w_2, \bar{w}_2)|0\rangle\langle 0|\mathcal{O}_i^\dagger(w_1, \bar{w}_1), \tag{2.2}$$

It is straightforward to check that the insertion points of the operators on the first sheet in the $w$ coordinate are as follows [6]

$$w_1 = i(a - it) - l, \qquad \bar{w}_1 = -i(a - it) - l,$$

$$w_2 = -i(a + it) - l, \qquad \bar{w}_2 = i(a + it) - l. \tag{2.3}$$

Next, by applying the replica trick [38], one can make $n$ copies, i.e. replicas, of the original plane $\mathbb{R}^2$ (See Figure 1). Then, one can write the difference $\Delta S_A^{(n)}$ between the n-th REE of the excited state and that of the vacuum state as follows [9–11]

$$\Delta S_A^{(n)} = S^{(n)}(\rho_A) - S^{(n)}(\rho_A^{(0)})$$

$$= \frac{1}{1-n} \log \left[ \frac{\text{Tr}\,(\rho_A)^n}{\text{Tr}\left(\rho_A^{(0)}\right)^n} \right]$$

$$= \frac{1}{1-n} \left[ \log\left(\frac{Z_n}{(Z_1)^n}\right) - \log\left(\frac{Z_{0n}}{(Z_{01})^n}\right) \right], \tag{2.4}$$

where $Z_n$ and $Z_{0n}$ are the partition functions on the n-sheeted Riemann surface $\Sigma_n$ for the excited

---

[6]Notice that we consider $a \pm it$ as a real number similar to refs. [9–11, 40].

and vacuum states, respectively. Moreover, one can verify that [9]

$$\frac{\text{Tr} \left( \rho_A \right)^n}{\text{Tr} \left( \rho_A^{(0)} \right)^n} = \frac{\langle \mathcal{O}_i^\dagger(w_1, \bar{w}_1) \mathcal{O}_i(w_2, \bar{w}_2) \cdots \mathcal{O}_i(w_{2n}, \bar{w}_{2n}) \rangle_{\Sigma_n}}{\left( \langle \mathcal{O}_i^\dagger(w_1, \bar{w}_1) \mathcal{O}_i(w_2, \bar{w}_2) \rangle_{\Sigma_1} \right)^n}, \tag{2.5}$$

where $\Sigma_1 = \mathbb{R}^2$ and the insertion points of the operators on the k-th sheet of $\Sigma_n$ are given by [12]

$$w_{2k+1} = e^{2\pi i k} w_1, \qquad\qquad w_{2k+2} = e^{2\pi i k} w_2. \tag{2.6}$$

In other words, by going from the $n = 1$ to $n = 2$ sheet, the phase of the $w$ coordinate changes by $2\pi i$. Next, by plugging eq. (2.5) into eq. (2.4), one arrives at [9–11]

$$\Delta S_A^{(n)} = \frac{1}{1-n} \left[ \log \langle \mathcal{O}_i^\dagger(w_1, \bar{w}_1) \mathcal{O}_i(w_2, \bar{w}_2) \cdots \mathcal{O}_i(w_{2n}, \bar{w}_{2n}) \rangle_{\Sigma_n} \right.$$

$$\left. -n \log \langle \mathcal{O}_i^\dagger(w_1, \bar{w}_1) \mathcal{O}_i(w_2, \bar{w}_2) \rangle_{\Sigma_1} \right]. \tag{2.7}$$

Therefore, one needs to calculate a $2n$-point function on $\Sigma_n$ and a two-point function on $\Sigma_1$. In the following, we consider the second REE. In this case, one has

$$\Delta S_A^{(2)} = -\log \left[ \frac{\langle \mathcal{O}_i^\dagger(w_1, \bar{w}_1) \mathcal{O}_i(w_2, \bar{w}_2) \mathcal{O}_i^\dagger(w_3, \bar{w}_3) \mathcal{O}_i(w_4, \bar{w}_4) \rangle_{\Sigma_2}}{\left( \mathcal{O}_i^\dagger(w_1, \bar{w}_1) \mathcal{O}_i(w_2, \bar{w}_2) \rangle_{\Sigma_1} \right)^2} \right]. \tag{2.8}$$

Notice that from eqs. (2.3) and (2.6), one can simply find the insertion points of the operators on the second sheet as follows

$$w_3 = (i(a - it) - l) e^{2\pi i}, \qquad\qquad w_4 = (-i(a + it) - l) e^{2\pi i}. \tag{2.9}$$

Next, by applying the following conformal transformation

$$z^n = \frac{w}{w - L}, \tag{2.10}$$

where $L$ is the length of the entangling region, one can map the n-sheeted Riemann surface $\Sigma_n$ with coordinate $w = x + i\tau$ to a complex plane $\Sigma_1 = \mathbb{C}$ with coordinate $z$. From eqs. (2.3), (2.9) and (2.10), the insertion points on $\mathbb{C}$ are as follows

$$z_1 = -z_3 = \sqrt{\frac{ia + t - l}{ia + t - l - L}}, \qquad\qquad z_2 = -z_4 = \sqrt{\frac{ia - t + l}{ia - t + l + L}}. \tag{2.11}$$

Now, we calculate the correlation functions in eq. (2.8). On $\Sigma_1 = \mathbb{R}^2$, in the $w$ coordinate, one can write the two-point function of a primary operator $\mathcal{O}_i$ as follows

$$\langle \mathcal{O}_i^\dagger(w_1, \bar{w}_1) \mathcal{O}_i(w_2, \bar{w}_2) \rangle_{\Sigma_1} = \frac{1}{|w_{12}|^{4\Delta_i}}, \tag{2.12}$$

where $w_{12} = w_1 - w_2$ and $\Delta_i$ is the conformal dimension [7] of the operator $\mathcal{O}_i$. On the other hand, from eq. (2.10), one has

$$w = \frac{Lz^2}{z^2 - 1},\tag{2.13}$$

and hence one arrives at

$$w_{12} = \frac{L(z_2^2 - z_1^2)}{(z_1^2 - 1)(z_2^2 - 1)}.\tag{2.14}$$

Then, by applying eq. (2.14), one can rewrite eq. (2.12) in terms of $z_{1,2}$ as follows

$$\langle \mathcal{O}_i^\dagger(w_1, \bar{w}_1) \mathcal{O}_i(w_2, \bar{w}_2) \rangle_{\Sigma_1} = \left| \frac{(z_1^2 - 1)(z_2^2 - 1)}{L(z_2^2 - z_1^2)} \right|^{4\Delta_i}.\tag{2.15}$$

Now, we want to calculate the $2n$-point functions of the primary operators on $\Sigma_n$. To do so, we use the fact that we are working with a free quantum field theory. Therefore, one can easily calculate the n-point function of the primary operator $e^{i\alpha\phi(z, \bar{z})}$ as follows [41] [8]

$$\langle e^{i\alpha_1\phi(z_1, \bar{z}_1)} e^{i\alpha_2\phi(z_2, \bar{z}_2)} \cdots e^{i\alpha_n\phi(z_n, \bar{z}_n)} \rangle_{\Sigma_1} = \prod_{i<j} |z_{ij}|^{2\alpha_i\alpha_j},\tag{2.16}$$

whenever the neutrality condition

$$\sum_{i=1}^{n} \alpha_i = 0,\tag{2.17}$$

is satisfied. Otherwise, the n-point function is zero. Now, by applying eq. (2.16) one can write the four-point function on $\Sigma_1$ in the $z$ coordinate as follows [10]

$$\langle \mathcal{O}_i^\dagger(z_1, \bar{z}_1) \mathcal{O}_i(z_2, \bar{z}_2) \mathcal{O}_i^\dagger(z_3, \bar{z}_3) \mathcal{O}_i(z_4, \bar{z}_4) \rangle_{\Sigma_1} = \frac{1}{|z_{13}z_{24}|^{4\Delta_i}} G_i(\eta, \bar{\eta}),\tag{2.18}$$

where $G_i(\eta, \bar{\eta})$ is a function of the cross ratios $\eta$ and $\bar{\eta}$ which are defined as follows [41]

$$\eta = \frac{z_{12}z_{34}}{z_{13}z_{24}} = -\frac{(z_1 - z_2)^2}{4z_1z_2},$$

$$\bar{\eta} = \frac{\bar{z}_{12}\bar{z}_{34}}{\bar{z}_{13}\bar{z}_{24}} = -\frac{(\bar{z}_1 - \bar{z}_2)^2}{4\bar{z}_1\bar{z}_2},\tag{2.19}$$

where $z_{ij} = z_i - z_j$. Furthermore, one has [41]

$$1 - \eta = \frac{z_{14}z_{23}}{z_{13}z_{24}} = \frac{(z_1 + z_2)^2}{4z_1z_2}.\tag{2.20}$$

---

[7]Here $\Delta_i = h_i = \bar{h}_i$, where $h_i$ and $\bar{h}_i$ are the holomorphic and anti-holomorphic conformal dimensions.

[8]We would like to thank Song He and Kento Watanabe for very helpful discussions regarding this calculation in ref. [10].

On the other hand, by applying eqs. (2.10) and (2.18), one can write the four-point function on $\Sigma_2$ in the $w$ coordinate as follows

$$
\begin{aligned}
\langle \prod_{l=1}^{2} \mathcal{O}_i^\dagger(w_{2l-1}, \bar{w}_{2l-1}) \mathcal{O}_i(w_{2l}, \bar{w}_{2l}) \rangle_{\Sigma_2} &= \prod_{j=1}^{4} \left| \frac{dw_j}{dz_j} \right|^{-2\Delta_i} \langle \mathcal{O}_i^\dagger(z_1, \bar{z}_1) \mathcal{O}_i(z_2, \bar{z}_2) \mathcal{O}_i^\dagger(z_3, \bar{z}_3) \mathcal{O}_i(z_4, \bar{z}_4) \rangle_{\Sigma_1} \\
&= \left| \frac{(z_1^2 - 1)^2 (z_2^2 - 1)^2}{4L^2 z_1 z_2} \right|^{4\Delta_i} \frac{1}{|z_{13} z_{24}|^{4\Delta_i}} G_i(\eta, \bar{\eta}) \\
&= \left| \frac{(z_1^2 - 1)(z_2^2 - 1)}{4L z_1 z_2} \right|^{8\Delta_i} G_i(\eta, \bar{\eta}).
\end{aligned} \tag{2.21}
$$

Then, from eqs. (2.15) and (2.21), one finds [10]

$$
\begin{aligned}
\frac{\text{Tr}\,(\rho_A)^2}{\text{Tr}\left(\rho_A^{(0)}\right)^2} &= \frac{\langle \mathcal{O}_i^\dagger(w_1, \bar{w}_1) \mathcal{O}_i(w_2, \bar{w}_2) \mathcal{O}_i^\dagger(w_3, \bar{w}_3) \mathcal{O}_i(w_4, \bar{w}_4) \rangle_{\Sigma_2}}{\left( \mathcal{O}_i^\dagger(w_1, \bar{w}_1) \mathcal{O}_i(w_2, \bar{w}_2) \rangle_{\Sigma_1} \right)^2} \\
&= \left| \frac{(z_2^2 - z_1^2)}{4 z_1 z_2} \right|^{8\Delta_i} G_i(\eta, \bar{\eta}) \\
&= |\eta(1 - \eta)|^{4\Delta_i} G_i(\eta, \bar{\eta}).
\end{aligned} \tag{2.22}
$$

where in the last line, we applied the following identities

$$
\eta = -\frac{(z_1 - z_2)^2}{4 z_1 z_2}, \qquad\qquad 1 - \eta = \frac{(z_1 + z_2)^2}{4 z_1 z_2}. \tag{2.23}
$$

Next, from eqs. (2.8) and (2.22), one obtains the second REE of the primary operator $\mathcal{O}_i$ as follows

$$
\Delta S_A^{(2)} = -\log \left[ |\eta(1 - \eta)|^{4\Delta_i} G_i(\eta, \bar{\eta}) \right]. \tag{2.24}
$$

In the following, we review the calculation of $\Delta S_A^{(2)}$ for two primary operators

$$
\mathcal{O}_1 = e^{\frac{i}{2}\phi}, \tag{2.25}
$$

$$
\mathcal{O}_2 = \frac{e^{-\frac{i}{2}\phi} + e^{\frac{i}{2}\phi}}{\sqrt{2}}, \tag{2.26}
$$

whose conformal dimensions are given by $\Delta_1 = \Delta_2 = \frac{1}{8}$.

### 2.1.1 $\quad \mathcal{O}_1$

For this operator, by applying eq. (2.16), one obtains

$$
\langle \mathcal{O}_1^\dagger(z_1, \bar{z}_1) \mathcal{O}_1(z_2, \bar{z}_2) \mathcal{O}_1^\dagger(z_3, \bar{z}_3) \mathcal{O}_1(z_4, \bar{z}_4) \rangle_{\Sigma_1} = \left| \frac{z_{13} z_{24}}{z_{12} z_{14} z_{23} z_{34}} \right|^{\frac{1}{2}}. \tag{2.27}
$$

Next, from eqs. (2.18) and (2.27), one simply find

$$
G_1(\eta, \bar{\eta}) = \frac{|z_{13} z_{24}|}{|z_{12} z_{14} z_{23} z_{34}|^{\frac{1}{2}}} = \frac{1}{\sqrt{|\eta(1 - \eta)|}}. \tag{2.28}
$$

At the end, by plugging eq. (2.28) into eq. (2.24), one arrives at [10]

$$\Delta S_A^{(2)} = -\log\left[\sqrt{|\eta(1-\eta)|}G_1(\eta,\bar\eta)\right]$$

$$= 0. \tag{2.29}$$

Therefore, the second REE for the excited state constructed by the operator $\mathcal{O}_1$, is the same as that for the vacuum state. This result can be interpreted by the quasi-particle picture. To do so, recall that one can decompose a scalar field in two-dimensions into its left-moving $\phi_L$ and right-moving $\phi_R$ modes, i.e. $\phi = \phi_L(t+x) + \phi_R(t-x)$. Having said this, it is easy to see that the operator $\mathcal{O}_1$ applied on the vacuum state, creates a separable state $|e^{\frac{i}{2}\phi_L}\rangle_L|e^{\frac{i}{2}\phi_R}\rangle_R$ [9]. In other words, the operator $\mathcal{O}_1$ inserted at $x = -l$, creates two quasi-particles moving at the speed of light to the left and right. Since, the quantum entanglement between these quasi-particles is zero, the REE of this excited sate is also zero [9, 10, 12].

### 2.1.2 $\mathcal{O}_2$

In this case, by applying eq. (2.16), one can find the four-point function of the operator $\mathcal{O}_2$ on $\Sigma_1$ as follows

$$\langle \mathcal{O}_2^\dagger(z_1,\bar z_1)\mathcal{O}_2(z_2,\bar z_2)\mathcal{O}_2^\dagger(z_3,\bar z_3)\mathcal{O}_2(z_4,\bar z_4)\rangle_{\Sigma_1} = \frac{1}{4}\Bigg[\langle -+-+\rangle_{\Sigma_1} + \langle -++-\rangle_{\Sigma_1}$$

$$+ \langle --++\rangle_{\Sigma_1} + \langle ++--\rangle_{\Sigma_1} + \langle +--+\rangle_{\Sigma_1} + \langle +-+-\rangle_{\Sigma_1}\Bigg], \tag{2.30}$$

where we defined

$$\langle -+-+\rangle_{\Sigma_1} = \langle e^{-\frac{i}{2}\phi(z_1,\bar z_1)}e^{+\frac{i}{2}\phi(z_2,\bar z_2)}e^{-\frac{i}{2}\phi(z_3,\bar z_3)}e^{+\frac{i}{2}\phi(z_4,\bar z_4)}\rangle_{\Sigma_1}, \tag{2.31}$$

and et cetera. It is straightforward to verify that

$$\langle -+-+\rangle_{\Sigma_1} = \langle +-+-\rangle_{\Sigma_1} = \left|\frac{z_{13}z_{24}}{z_{12}z_{14}z_{23}z_{34}}\right|^{\frac{1}{2}} = \frac{1}{|z_{13}z_{24}|^{\frac{1}{2}}}\frac{1}{\sqrt{|\eta(1-\eta)|}},$$

$$\langle -++-\rangle_{\Sigma_1} = \langle +--+\rangle_{\Sigma_1} = \left|\frac{z_{14}z_{23}}{z_{12}z_{13}z_{24}z_{34}}\right|^{\frac{1}{2}} = \frac{1}{|z_{13}z_{24}|^{\frac{1}{2}}}\frac{|1-\eta|}{\sqrt{|\eta(1-\eta)|}},$$

$$\langle --++\rangle_{\Sigma_1} = \langle ++--\rangle_{\Sigma_1} = \left|\frac{z_{12}z_{34}}{z_{13}z_{14}z_{23}z_{24}}\right|^{\frac{1}{2}} = \frac{1}{|z_{13}z_{24}|^{\frac{1}{2}}}\frac{|\eta|}{\sqrt{|\eta(1-\eta)|}}. \tag{2.32}$$

Next, by plugging eq. (2.32) into eq. (2.30) and applying eq. (2.18), one finds [10]

$$G_2(\eta,\bar\eta) = \frac{1}{2\sqrt{|\eta(1-\eta)|}}\left(1 + |\eta| + |1-\eta|\right). \tag{2.33}$$

Then, from eqs. (2.24) and (2.33), one has [10]

$$\Delta S_A^{(2)} = -\log\left[\sqrt{|\eta(1-\eta)|}G_2(\eta,\bar\eta)\right]$$

$$= \log \left[ \frac{2}{1 + |\eta| + |1 - \eta|} \right].$$
(2.34)

It is straightforward to check that when $a \to 0$, for $0 < t < l$ and $t > l + L$, one has

$$\eta = \frac{L^2 a^2}{4(l - t)^2 (L + l - t)^2} \to 0.$$
(2.35)

Therefore, for $0 < t < l$ and $t > l + L$, to the zeroth order in $a$, one obtains [10]

$$\Delta S_A^{(2)} = 0.$$
(2.36)

On the other hand, for $l < t < l + L$, one has

$$\eta = 1 - \frac{L^2 a^2}{4(l - t)^2 (L + l - t)^2} \to 1.$$
(2.37)

Thus, for $l < t < l + L$, to the zeroth order in $a$, one arrives at [10]

$$\Delta S_A^{(2)} = \log 2.$$
(2.38)

Putting everything together, one can write

$$\Delta S_A^{(2)} = \begin{cases} 0 & 0 < t < l, \\ \log 2, & l < t < l + L, \\ 0 & t > l + L. \end{cases}$$
(2.39)

Notice that it is always real valued and positive. Moreover, its late time value is zero [10]. We plotted $\Delta S_A^{(2)}$ in Figure 2. It should be pointed out that eq. (2.39) can be explained by the quasi-particle picture. To do so, notice that the operator $\mathcal{O}_2$ creates an EPR state as follows [9, 11]

$$|\mathcal{O}_2\rangle|0\rangle = \frac{1}{\sqrt{2}} \left( |e^{-\frac{i}{2}\phi_L}\rangle_L |e^{-\frac{i}{2}\phi_R}\rangle_R + |e^{\frac{i}{2}\phi_L}\rangle_L |e^{\frac{i}{2}\phi_R}\rangle_R \right).$$
(2.40)

In other words, the operator $\mathcal{O}_2$ inserted at $x = -l$, creates a left moving and a right moving quasi-particle moving at the speed of light. At very early times, the two particles are inside the region $B$, i.e. the complement of the entangling region $A$, and hence, the quantum entanglement between them is zero. As time elapses, the left moving particle keeps moving inside $B$. However, the right moving particle enters the region $A$, and hence the REE is $\log 2$ for every $n$ [9–12]. At late times, the left moving particle goes outside $A$. Since the two quasi-particles are now inside $B$, there is no quantum entanglement between them. Thus, the REE becomes zero again [10, 12].

## 2.2 Semi-Infinite Interval

When the entangling region is a semi-infinite interval, i.e. $A \in [0, \infty]$, the calculation is the same as in the previous section. However, one should replace the conformal map in eq. (2.10) with the following transformation [10] (See also ref. [29])

$$z^n = w.$$
(2.41)

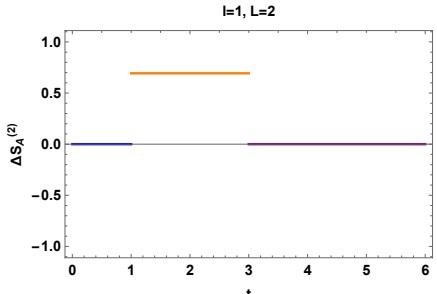

Figure 2: $\Delta S_A^{(2)}$ as a function of $t$ for $l = 1$, $L = 2$. The values of $\Delta S_A^{(2)}$ in the time intervals $0 < t < l$, $l < t < l + L$ and $t > l + L$ are shown in blue, orange and purple, respectively. At late times it saturates to zero.

Next, from eq. (2.3), one obtains the insertion points of the operators as follows

$$z_1 = -z_3 = \sqrt{ia + t - l}, \qquad z_2 = -z_4 = \sqrt{-ia + t - l}. \qquad (2.42)$$

Moreover, eqs. (2.15) and (2.21) are replaced by

$$\langle \mathcal{O}_i^\dagger(w_1, \bar{w}_1) \mathcal{O}_i(w_2, \bar{w}_2) \rangle_{\Sigma_1} = \frac{1}{\left| z_2^2 - z_1^2 \right|^{4\Delta_i}}, \qquad (2.43)$$

and

$$\langle \mathcal{O}_i^\dagger(w_1, \bar{w}_1) \mathcal{O}_i(w_2, \bar{w}_2) \mathcal{O}_i^\dagger(w_3, \bar{w}_3) \mathcal{O}_i(w_4, \bar{w}_4) \rangle_{\Sigma_2} = \frac{1}{\left| 4z_1 z_2 \right|^{8\Delta_i}} G_i(\eta, \bar{\eta}), \qquad (2.44)$$

respectively. Next, by applying eqs. (2.8), (2.43) and (2.44), one can conclude that $\Delta S_A^{(2)}$ is given by eq. (2.24) where the functions $G_i(\eta, \bar{\eta})$ are defined by eqs. (2.28) and (2.33).

### 2.2.1 $\mathcal{O}_1$

As mentioned above, eq. (2.29) is still valid for the operator $\mathcal{O}_1$, and hence one has $\Delta S_A^{(2)} = 0$.

### 2.2.2 $\mathcal{O}_2$

From eq. (2.42), one can simply find the cross rations. It is straightforward to show that in the zero cutoff limit, i.e. $a \to 0$, when $t < l$, one has

$$\eta = 1 - \frac{a^2}{4(l-t)^2} \to 1. \qquad (2.45)$$

Moreover, in this limit for $t > l$, one obtains

$$\eta = \frac{a^2}{4(l-t)^2} \to 0. \qquad (2.46)$$

Next, by plugging eqs. (2.45) and (2.46), into eq. (2.34), one obtains [9]

$$\Delta S_A^{(2)} = \begin{cases} 0 & 0 < t < l, \\ \log 2, & t > l. \end{cases} \qquad (2.47)$$

# 3   The Second Pseudo Rényi Entanglement Entropy For $CFT_2$

In this section, we calculate the second PREE for two different locally excited states

$$|\psi\rangle = e^{-aH}\mathcal{O}_i(x=-l)|0\rangle,$$

$$|\phi\rangle = e^{-iHt}e^{-a'H}\mathcal{O}_i(x=-l)|0\rangle, \tag{3.1}$$

in a two-dimensional CFT. Here $a$ and $a'$ are two different UV cutoffs, $|0\rangle$ is the vacuum state and $\mathcal{O}_i(x)$ is a primary operator. For $a' = a$, $|\phi\rangle$ is the time evolution of $|\psi\rangle$. In the following, we consider operators $\mathcal{O}_{1,2}$ introduced in eqs. (2.25) and (2.26) as well as the following operator

$$\mathcal{O}_3 = \frac{\cos\theta e^{-\frac{i}{2}\phi} + e^{\frac{i}{2}\phi}}{\sqrt{2}}, \tag{3.2}$$

and calculate the PREE for the two states given in eq. (3.1). The transition matrix $\tau^{\psi|\phi}$ is simply given by

$$\tau^{\psi|\phi} = |\psi\rangle\langle\phi|$$

$$= e^{-aH}\mathcal{O}_i(x=-l)|0\rangle\langle0|\mathcal{O}_i^\dagger(x=-l)e^{-a'H}e^{iHt},$$

$$= \mathcal{O}_i(w_2, \bar{w}_2)|0\rangle\langle0|\mathcal{O}_i^\dagger(w_1, \bar{w}_1), \tag{3.3}$$

where $w = x + i\tau$ and $\tau = -it$ is the Euclidean time. Next, similar to the case for the REE of the locally excited states, i.e. eq. (2.7), one can prove that [20]

$$\Delta S_A^{(n)} = S(\tau_A) - S(\rho_A^{(0)})$$

$$= \frac{1}{1-n}\log\left[\frac{\text{Tr}\,(\tau_A)^n}{\text{Tr}\left(\rho_A^{(0)}\right)^n}\right]$$

$$= \frac{1}{1-n}\left[\log\langle\mathcal{O}_a^\dagger(w_1, \bar{w}_1)\mathcal{O}_a(w_2, \bar{w}_2)\cdots\mathcal{O}_a^\dagger(w_{2n}, \bar{w}_{2n})\rangle_{\Sigma_n}\right.$$

$$\left. -n\log\langle\mathcal{O}_a^\dagger(w_1, \bar{w}_1)\mathcal{O}_a(w_2, \bar{w}_2)\rangle_{\Sigma_1}\right], \tag{3.4}$$

where $S(\rho_A^{(0)})$ is the n-th REE of the vacuum state. Notice that the transition matrix in eq. (3.3) is similar to the density matrix in eq. (2.2). However, as we will see in the following, the insertion points of the operators in the calculation of the second REE and PREE are different (See eqs. (2.3) and (3.5)). Therefore, for $n = 2$, the calculation of the PREE is the same as that for the second REE in the previous section. In other words, one can still apply eqs. (2.29) and (2.34) for the operators $\mathcal{O}_1$ and $\mathcal{O}_2$, respectively. However, since the insertion points are not the same, the cross ratios are different. In the following, we calculate $\Delta S_A^{(2)}$ for finite and semi-infinite intervals, respectively.

## 3.1 Finite Interval

In this case, the insertion points of the operators on the first sheet of $\Sigma_2$ are given by

$$w_1 = ia' + t - l, \qquad\qquad \bar{w}_1 = -ia' - t - l,$$

$$w_2 = -ia - l, \qquad\qquad \bar{w}_2 = ia - l. \tag{3.5}$$

Now, one can simply find the insertion points on the complex plane $\mathbb{C}$ as follows

$$z_1 = -z_3 = \sqrt{\frac{w_1}{w_1 - L}} = \sqrt{\frac{ia' + t - l}{ia' + t - l - L}},$$

$$z_2 = -z_4 = \sqrt{\frac{w_2}{w_2 - L}} = \sqrt{\frac{ia + l}{ia + l + L}}. \tag{3.6}$$

Moreover, the cross ratios $\eta$ and $\bar{\eta}$ are given by

$$\eta = -\frac{z_{12}^2}{4z_1 z_2}, \qquad\qquad \bar{\eta} = -\frac{\bar{z}_{12}^2}{4\bar{z}_1 \bar{z}_2}. \tag{3.7}$$

### 3.1.1 $\mathcal{O}_1$

As mentioned above, one can show that eq. (2.29) is valid, and hence one has $\Delta S_A^{(2)} = 0$. In other words, the second PREE is equal to the EE of the vacuum state.

### 3.1.2 $\mathcal{O}_2$

In this case, one can show that eq. (2.34) holds again, and hence the second PREE is simply given by

$$\Delta S_A^{(2)} = \log\left(\frac{2}{1 + |\eta| + |1 - \eta|}\right). \tag{3.8}$$

Next, by plugging eqs. (3.6) and (3.7) into eq. (3.8), one obtains

$$\Delta S_A^{(2)} = \log\left(\frac{8}{4 + A_- + A_+}\right), \tag{3.9}$$

where

$$A_{\pm} = \left(\frac{\left(a^2 + (l+L)^2\right)\left((l+L)^2 - (ia'+t)^2\right)}{(a^2 + l^2)(l^2 - (ia'+t)^2)}\right)^{\frac{1}{4}} \left(\sqrt{\frac{ia+l}{ia+l+L}} \pm \sqrt{\frac{-ia'+l-t}{-ia'+l+L-t}}\right)$$

$$\times \left(\sqrt{\frac{-ia+l}{-ia+l+L}} \pm \sqrt{\frac{ia'+l+t}{ia'+l+L+t}}\right). \tag{3.10}$$

Moreover, at late times, i.e. $t \to \infty$, $\Delta S_A^{(2)}$ saturates and approaches a constant value

$$\Delta S_A^{(2)} = \log\left(\frac{8}{4 + A_-^{\infty} + A_+^{\infty}}\right), \tag{3.11}$$

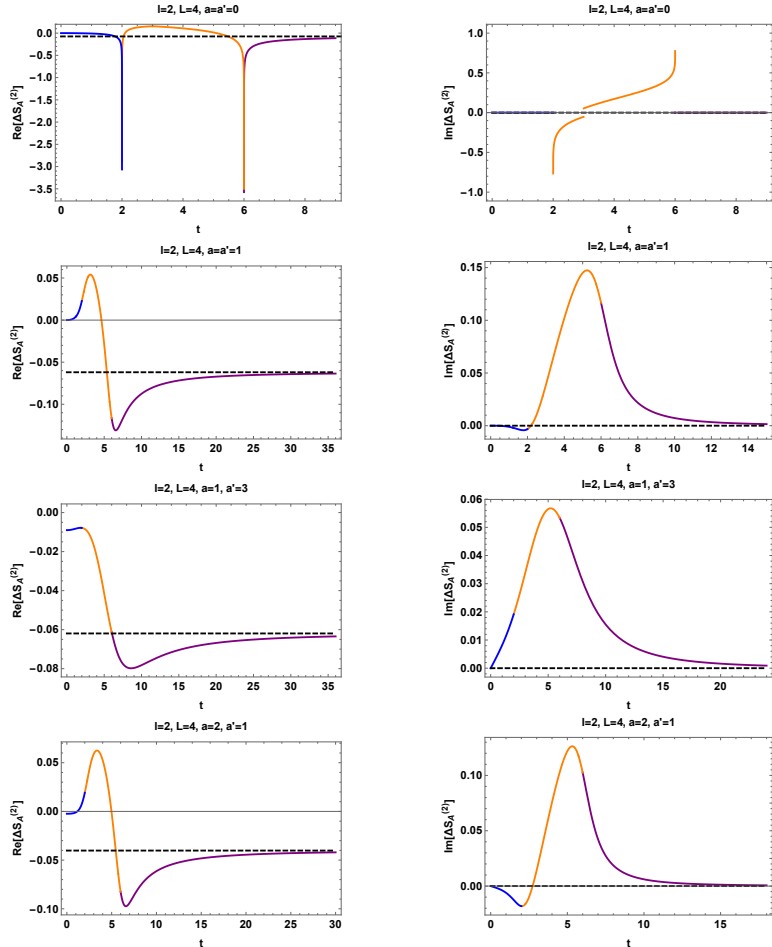

Figure 3: $\Delta S_A^{(2)}$ as a function of $t$ for the operator $\mathcal{O}_2$, $l = 2$, $L = 4$ and *First Row)* $a = a' = 0$ *Second Row)* $a = a' = 1$ *Third Row)* $a = 1, a' = 3$ *Fourth Row)* $a = 2, a' = 1$. The values of $\Delta S_A^{(2)}$ in the time intervals $0 < t < l$, $l < t < l + L$ and $t > l + L$ are shown in blue, orange and purple, respectively. At late times, it saturates to a constant real value given by eq. (3.11) which is shown by the dashed black line.

where

$$A_\pm^\infty = \left( \frac{a^2 + (l+L)^2}{a^2 + l^2} \right)^{\frac{1}{4}} \left( 1 \pm \sqrt{\frac{ia+l}{ia+l+L}} \right) \left( 1 \pm \sqrt{\frac{-ia+l}{-ia+l+L}} \right). \tag{3.12}$$

In Figure 3, we plotted the real and imaginary parts of the PREE $\Delta S_A^{(2)}$ as a function of $t$. It is observed that $\Delta S_A^{(2)}$ is always a complex number which is in contrast with the case for the REE of the state $|\phi\rangle$ (See Figures 2 and 4). It is observed that both the real and imaginary parts of the PREE can be either positive or negative. This behavior is in contrast with that of the second REE for an excited state studied in ref. [9] which is always real and it is either zero or positive (See Figure 4). Moreover, at late times $\Delta S_A^{(2)}$ saturates to a value given by eq. (3.11). For $a = a' = 0$, the real part has two sharp troughs (valleys) (See the first row in Figure 3). However, for non-zero cutoffs, the curves are smooth (See the second, third and fourth rows in Figure 3).

On the other hand, it is interesting to calculate the time evolution of the second REE of the locally

excited states $|\psi\rangle$ and $|\phi\rangle$. For the state $|\psi\rangle$, the density matrix is simply given by

$$\rho_\psi(t) = e^{-aH}\mathcal{O}_i(x=-l)|0\rangle\langle 0|\mathcal{O}_i^\dagger(x=-l)e^{-aH} \tag{3.13}$$

Therefore, one has

$$z_1 = -z_3 = \sqrt{\frac{ia-l}{ia-l-L}}, \qquad\qquad z_2 = -z_4 = \sqrt{\frac{ia+l}{ia+l+L}}. \tag{3.14}$$

In this case, the second REE is zero forever. [9] On the other hand, for the state $|\phi\rangle$, the density matrix is given by

$$\rho_\phi(t) = e^{-iHt}e^{-a'H}\mathcal{O}_i(x=-l)|0\rangle\langle 0|\mathcal{O}_i^\dagger(x=-l)e^{-a'H}e^{iHt}, \tag{3.15}$$

and hence one arrives at

$$z_1 = -z_3 = \sqrt{\frac{ia'+t-l}{ia'+t-l-L}},$$

$$z_2 = -z_4 = \sqrt{\frac{ia'-t+l}{ia'-t+l+L}}. \tag{3.16}$$

In Figure 4, we plotted the second REE $\Delta S_A^{(2)}$ for $|\phi\rangle$ and its time derivative as a function of $t$. It is observed that $\Delta S_A^{(2)}$ is always real and depends on $t$. Moreover, for $a'=0$, the curve is not continuous. However, for a finite cutoff, the curve is a smooth function of $t$.

It should be pointed out that the PREE was studied for a *global* quench in ref. [24] and it was observed that the PREE is a complex number. Moreover, the behavior of the real part of the PREE was similar to the REE of the states $|\phi\rangle$ and $|\psi\rangle$. On the other hand, the behavior of the imaginary part of the PREE was similar to that of the time derivative of the REE (See Figure 21 in ref. [24]). Comparison of Figures 3 and 4, shows that the real part of the second PREE does not behave the same as the REE of the state $|\phi\rangle$. On the other hand, the imaginary part of the second PREE does not behave the same as the time derivative of the REE of the state $|\phi\rangle$. Notice that here we considered a *local* quench. Moreover, in Figure 3, we did not plot $S^{(2)}(\tau_A)$, but we plotted the difference between $S^{(2)}(\tau_A)$ and the second REE, i.e. $S^{(2)}(\rho_A^{(0)})$, of the vacuum state. Furthermore, in Figure 4, we plotted the difference of the REE of the exited state and that of the vacuum. In other words, in both Figures, we subtracted the second REE of the vacuum state from the corresponding quantities. Therefore, it might not be surprising that we do not observe the behavior reported in ref. [24]. [10]

On the other hand, when the entangling region is a finite interval $A \in [x_l, x_r]$, eq. (3.6) are replaced by the following equations

$$z_1 = -z_3 = \sqrt{\frac{w_1-x_l}{w_1-x_r}} = \sqrt{\frac{ia'+t-l-x_l}{ia'+t-l-x_r}},$$

$$z_2 = -z_4 = \sqrt{\frac{w_2-x_l}{w_2-x_r}} = \sqrt{\frac{ia+l+x_l}{ia+l+x_r}}. \tag{3.17}$$

---

[9] The state $|\phi\rangle$ at $t=0$, is the same as $|\psi\rangle$. Therefore, $\Delta S_A^{(2)}$ for $|\psi\rangle$ has to be equal to that of $|\phi\rangle$ at $t=0$, which is zero.

[10] We would like to thank Ali Mollabashi very much for very useful discussions on this regard.

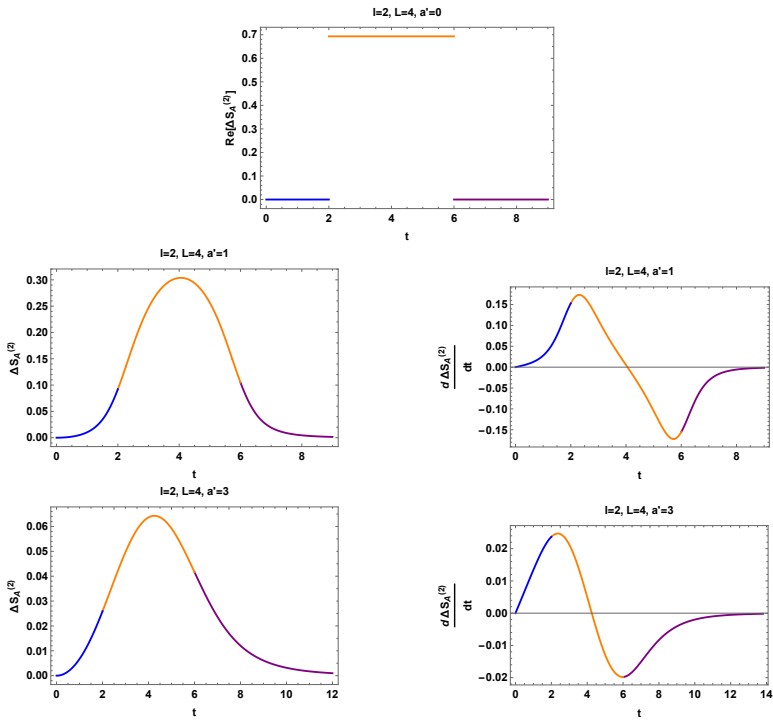

Figure 4: The second REE of the state $|\phi\rangle$, i.e. $\Delta S_A^{(2)}$, as a function of $t$ for $l = 2$, $L = 4$ and *Left*) $a' = 0$ *Middle*) $a' = 1$ *Right*) $a' = 3$. The values of $\Delta S_A^{(2)}$ in the time intervals $0 < t < l$, $l < t < l + L$ and $t > l + L$ are shown in blue, orange and purple, respectively.

In this case, the center of the interval $A$ is located at

$$x_m = \frac{x_l + x_r}{2}. \tag{3.18}$$

In Figures 5, 6, 7 and 8, we plotted the real and imaginary parts of $\Delta S_A^{(2)}$ as a function of $x_m$ for different values of $t$. From figure 5, it is evident that for $t = 0$, when the endpoints $x_{l,r}$ of the entangling region get closer to the insertion point of the operator $\mathcal{O}_2$ at $x = -l$, the second PREE is very suppressed such that there are two troughs. The left trough happens when $x_r$ coincides with the insertion point at $x = -l$ and the right trough happens when $x_l$ coincides with $x = -l$. It should be pointed out that this suppression was observed previously for the second PREE between two different excited states in ref. [20] and it was interpreted as a result of *entanglement swapping*.

In Figure 6, one has $a < a'$. When the endpoints of the interval coincide the insertion point at $x = -l$, the real part of $\Delta S_A^{(2)}$ becomes suppressed very much, such that there are two troughs. As time elapses, the depth of the troughs becomes larger in time. Moreover, when the endpoints of the interval coincide the insertion point, the imaginary part of $\Delta S_A^{(2)}$ increases very much, such that there are two peaks. As time passes, the heights of the peaks increase with time.

In Figure 7, one has $a > a'$. When the endpoints of the interval coincide the insertion point at $x = -l$, both the real and imaginary parts of $\Delta S_A^{(2)}$ are suppressed at early times. However, as time elapses they increase such that two peaks appear. Moreover, the heights of the peaks increase in time.

In Figure 8, one has $a = a'$. In this case, when the endpoints of the interval coincides the insertion point at $x = -l$, both the real and imaginary parts peak. The heights of the peaks grows in time. Moreover, both the real and imaginary parts of the second PREE in Figures 5, 6, 7 and 8 are symmetric around the insertion point of the operator $\mathcal{O}_2$ at $x = -l$.

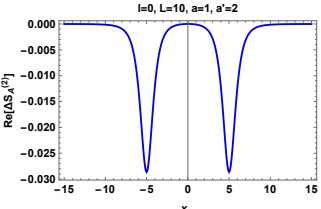
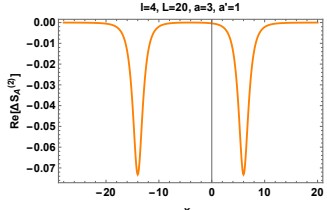

Figure 5: $\Delta S_A^{(2)}$ as a function of $x_m$ for the operator $\mathcal{O}_2$, $t = 0$ and *Left) $l = 0$, $L = 10$, $a = 1$, $a' = 2$ Right) $l = 4$, $L = 20$, $a = 3$, $a' = 1$*. This Figure is the same as Figure 9 in ref. [20]. It should be pointed out that $\Delta S_A^{(2)}$ is real at $t = 0$. Moreover, the plots are symmetric around the insertion point of the operator $\mathcal{O}_2$ at $t = 0$.

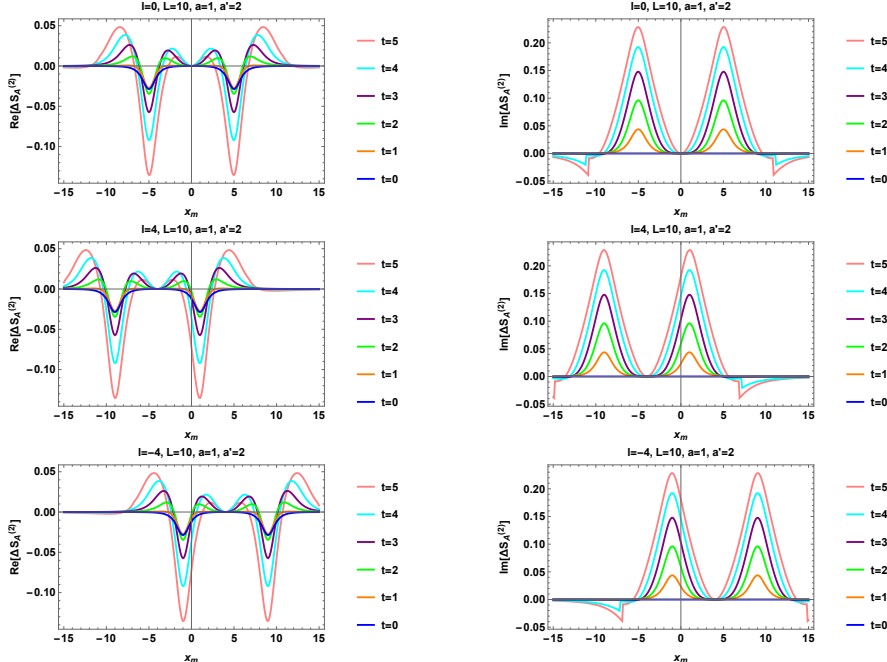

Figure 6: $\Delta S_A^{(2)}$ as a function of $x_m$ for the operator $\mathcal{O}_2$, $L = 10$, $a = 1$, $a' = 2$ *Top Row) $l = 0$ Middle Row) $l = 4$ Down Row) $l = -4$*.

### 3.1.3 $\mathcal{O}_3$

The conformal dimension of the operator $\mathcal{O}_3$ is $\Delta_3 = \frac{1}{8}$. In this case, one can calculate the two-point function of this operator on $\Sigma_1$ as follows [11]

$$
\begin{aligned}
\langle \mathcal{O}_3^\dagger(w_1, \bar{w}_1) \mathcal{O}_3(w_2, \bar{w}_2) \rangle_{\Sigma_1} &= \frac{1}{2} \left[ \langle -+ \rangle + \cos^2\theta \langle +- \rangle \right] \\
&= \frac{(1 + \cos^2\theta)}{2} \frac{1}{|w_{12}|^{\frac{1}{2}}} \\
&= \frac{(1 + \cos^2\theta)}{2} \left| \frac{(z_1^2 - 1)(z_2^2 - 1)}{L(z_2^2 - z_1^2)} \right|^{\frac{1}{2}}.
\end{aligned}
\tag{3.19}
$$

---

[11]One can also consider the operator $\mathcal{O}_4 = \frac{e^{i\theta} e^{-\frac{i}{2}\phi} + e^{\frac{i}{2}\phi}}{\sqrt{2}}$. It is straightforward to verify that the phase $e^{i\theta}$ is canceled in both the two-point and four-point functions. Therefore, for this operator the function $G_4(\eta, \bar{\eta})$ and $\Delta S_A^{(2)}$ are given by eqs. (2.33) and (2.34), respectively. For this reason, we discarded the operator $\mathcal{O}_4$ and work with the operator $\mathcal{O}_3$ introduced in eq. (3.2).

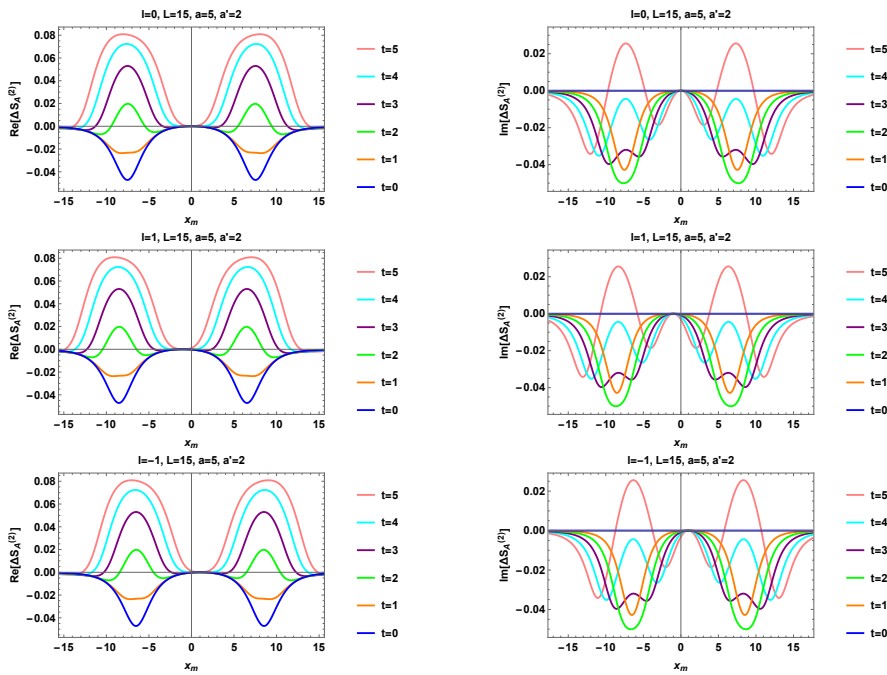

Figure 7: $\Delta S_A^{(2)}$ as a function of $x_m$ for the operator $\mathcal{O}_2$, $L = 15$, $a = 5$, $a' = 2$ *Top Row*) $l = 0$ *Middle Row*) $l = 1$ *Down Row*) $l = -1$.

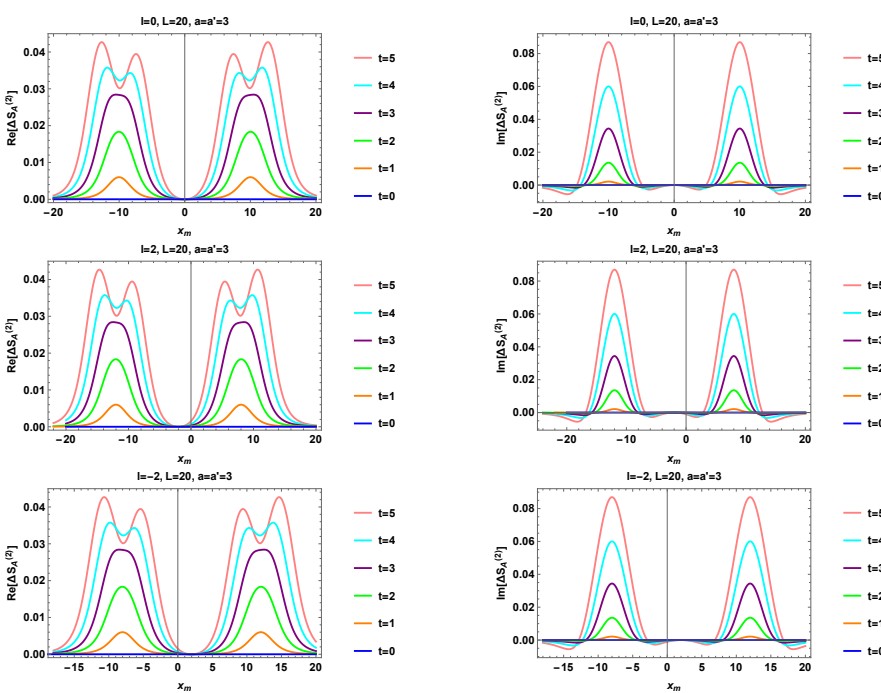

Figure 8: $\Delta S_A^{(2)}$ as a function of $x_m$ for the operator $\mathcal{O}_2$, $L = 20$, $a = a' = 3$ *Top Row*) $l = 0$ *Middle Row*) $l = 2$ *Down Row*) $l = -2$.

On the other hand, the four-point function in the z-coordinate is given by

$$\langle \mathcal{O}_3^\dagger(z_1, \bar{z}_1)\mathcal{O}_3(z_2, \bar{z}_2)\,\mathcal{O}_3^\dagger(z_3, \bar{z}_3)\mathcal{O}_3(z_4, \bar{z}_4)\rangle_{\Sigma_1} = \frac{1}{4}\left[\langle -+-+\rangle_{\Sigma_1} + \cos^4\theta\langle +--+\rangle_{\Sigma_1}\right.$$

$$+ \cos^2\theta\left(\langle -++-\rangle_{\Sigma_1} + \langle --++\rangle_{\Sigma_1} + \langle ++--\rangle_{\Sigma_1} + \langle +--+\rangle_{\Sigma_1}\right)\right]$$

$$= \frac{1}{4\sqrt{|z_{13}z_{24}\,\eta(1-\eta)|}}\left[(1+\cos^4\theta) + 2\cos^2\theta\left(|\eta| + |1-\eta|\right)\right], \tag{3.20}$$

where in the last equality we applied eq. (2.32). From the above equation, one simply finds

$$G_3(\eta, \bar{\eta}) = \frac{1}{4\sqrt{|\eta(1-\eta)|}}\left[(1+\cos^4\theta) + 2\cos^2\theta\left(|\eta| + |1-\eta|\right)\right]. \tag{3.21}$$

Next, from eqs. (2.21) and (3.21), one can write the four-point function on $\Sigma_2$ as follows

$$\langle \mathcal{O}_3^\dagger(w_1, \bar{w}_1)\mathcal{O}_3(w_2, \bar{w}_2)\mathcal{O}_3^\dagger(w_3, \bar{w}_3)\mathcal{O}_3(w_4, \bar{w}_4)\rangle_{\Sigma_2} = \left|\frac{(z_1^2-1)(z_2^2-1)}{4Lz_1z_2}\right|G_3(\eta, \bar{\eta}). \tag{3.22}$$

By plugging eqs. (3.19) and (3.22) into eq. (3.4), one has

$$\frac{\text{Tr}\,(\tau_A)^2}{\text{Tr}\left(\rho_A^{(0)}\right)^2} = \frac{\langle \mathcal{O}_3^\dagger(w_1, \bar{w}_1)\mathcal{O}_3(w_2, \bar{w}_2)\mathcal{O}_3^\dagger(w_3, \bar{w}_3)\mathcal{O}_3(w_4, \bar{w}_4)\rangle_{\Sigma_2}}{\left(\mathcal{O}_3^\dagger(w_1, \bar{w}_1)\mathcal{O}_3(w_2, \bar{w}_2)\rangle_{\Sigma_1}\right)^2}$$

$$= \frac{4}{(1+\cos^2\theta)^2}\left|\frac{(z_2^2-z_1^2)}{4z_1z_2}\right|G_3(\eta, \bar{\eta})$$

$$= \frac{4}{(1+\cos^2\theta)^2}\sqrt{|\eta(1-\eta)|}G_3(\eta, \bar{\eta}). \tag{3.23}$$

Then, form eqs. (3.4) and (3.23), one obtains

$$\Delta S_A^{(2)} = \log\left[\frac{\left(1+\cos^2\theta\right)^2}{(1+\cos^4\theta) + 2\cos^2\theta\left(|\eta| + |1-\eta|\right)}\right] \tag{3.24}$$

$$= \log\left[\frac{2\left(1+\cos^2\theta\right)^2}{2\left(1+\cos^4\theta\right) + \cos^2\theta\left(A_- + A_+\right)}\right], \tag{3.25}$$

where $A_\pm$ are given by eq. (3.10). Notice that for $\theta = \frac{\pi}{2}$ and $\theta = 0$, it reduces to eqs. (2.29) and (2.34), respectively. Moreover, it is an even function of $\theta$. In Figure 9, we plotted $\Delta S_A^{(2)}$ as a function of $\theta$ for different values of the cutoffs and times. Notice that for $\theta = \frac{\pi}{2}$, both $|\psi\rangle$ and $|\phi\rangle$ are separable, and hence it is not surprising that $\Delta S_A^{(2)}$ is zero.

## 3.2   Semi-Infinite Interval

In this section, we consider an entangling region which is a semi-infinite interval. From eqs. (2.41) and (3.5), one has

$$z_1 = -z_3 = \sqrt{ia' + t - l}, \qquad\qquad z_2 = -z_4 = i\sqrt{ia + l}. \tag{3.26}$$

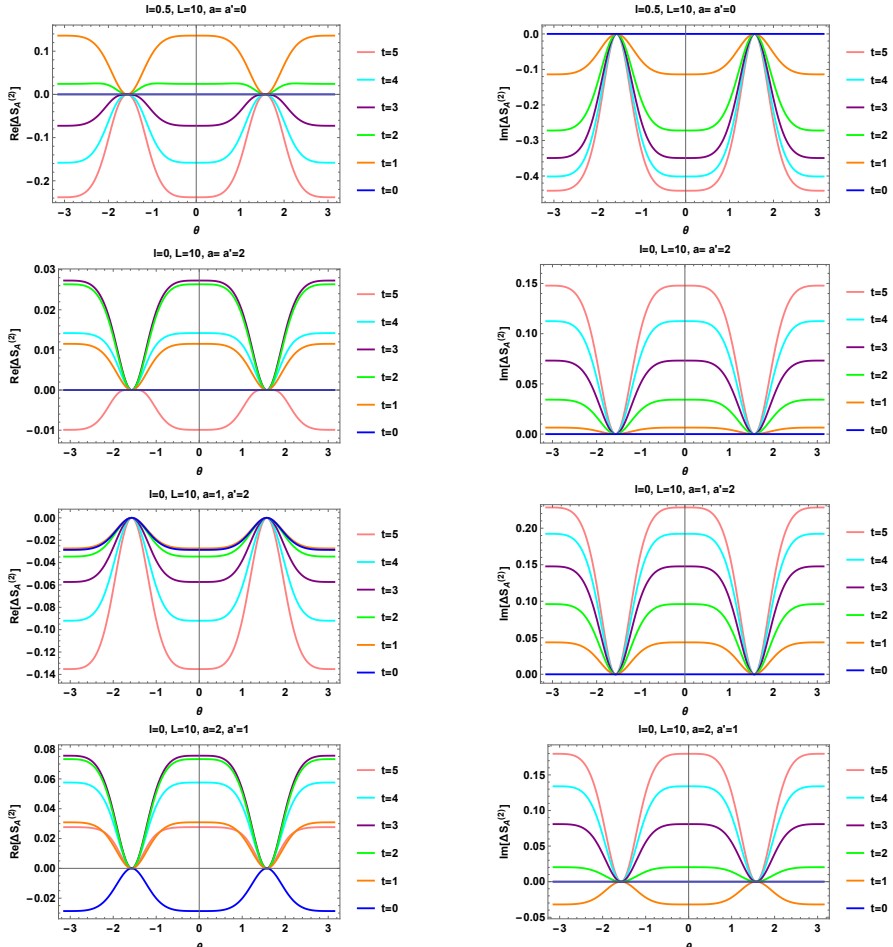

Figure 9: The real and imaginary parts of $\Delta S_A^{(2)}$ for the operator $\mathcal{O}_3$ as a function of $\theta$ for: *First Row)* $a = a' = 0$, $l = 0.5$ *Second Row)* $a = a' = 2$, $l = 0$ *Third Row)* $a = 1$, $a' = 2$, $l = 0$ *Fourth Row)* $a = 2$, $a' = 1$, $l = 0$. In all of the figures, we set $L = 10$.

### 3.2.1 $\mathcal{O}_1$

For the operator $\mathcal{O}_1$, one can show that eq. (2.29) is valid, and hence the second PREE is zero again. Therefore, in the following we consider operators $\mathcal{O}_{2,3}$.

### 3.2.2 $\mathcal{O}_2$

For the operator $\mathcal{O}_2$, by plugging eq. (3.26) into eq. (3.8), one obtains

$$\Delta S_A^{(2)} = \log\left(\frac{8}{4 + B_- + B_+}\right), \tag{3.27}$$

where

$$B_\pm = \frac{\left(\sqrt{ia + l} \pm \sqrt{-ia' + l - t}\right)\left(\sqrt{-ia + l} \pm \sqrt{ia' + l + t}\right)}{\left((a^2 + l^2)\left(l^2 - (ia' + t)^2\right)\right)^{\frac{1}{4}}}. \tag{3.28}$$

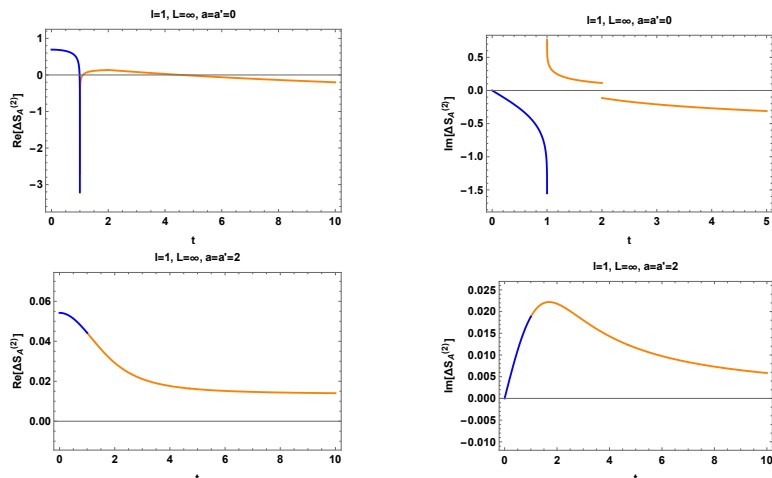

Figure 10: $\Delta S_A^{(2)}$ as a function of $t$ for the operator $\mathcal{O}_2$, $L = \infty$ and *First Row*) $l = 2$ and $a = a' = 0$ *Second Row*) $l = 1$ and $a = a' = 2$. The values of $\Delta S_A^{(2)}$ in the time intervals $0 < t < l$ and $t > l$ are shown in blue and orange, respectively.

In the limits $a \to 0$ and $a' \to 0$, one has

$$B_\pm \to B_\pm^{(0)} = \frac{\left(\sqrt{l} \pm \sqrt{l-t}\right)\left(\sqrt{l} \pm \sqrt{t+l}\right)}{\sqrt{l}\,(l^2 - t^2)^{\frac{1}{4}}}. \tag{3.29}$$

In Figure 10, we plotted $\Delta S_A^{(2)}$ as a function of $t$. For $a' = a = 0$, $\Delta S_A^{(2)}$ is not a smooth function at $t = l$. However, for non-zero cutoffs, it is a smooth function at $t = l$.

### 3.2.3 $\mathcal{O}_3$

For the operator $\mathcal{O}_3$, by plugging eq. (3.26) into eq. (3.24), one has

$$\Delta S_A^{(2)} = \log\left[\frac{2\left(1 + \cos^2\theta\right)^2}{2\left(1 + \cos^4\theta\right) + \cos^2\theta\left(B_- + B_+\right)}\right], \tag{3.30}$$

where $B_\pm$ are given by eq. (3.28). In Figure 11, we plotted $\Delta S_A^{(2)}$ as a function of $t$ for $a' = a$ and different values of $\theta$. Since it is a function of $\cos^2\theta$, we restricted ourselves to the case of $0 < \theta < \frac{\pi}{2}$. For zero cutoffs, the real and imaginary parts of $\Delta S_A^{(2)}$ are not smooth functions of $t$. On the other hand, for nonzero cutoffs, it is observed that the real and imaginary parts of $\Delta S_A^{(2)}$ are decreasing functions of $\theta$.

## 4 The Third Pseudo Rényi Entanglement Entropy

In this section, we calculate the third PREE, i.e. $\Delta S_A^{(3)}$, for the operators $\mathcal{O}_{1,2,3}$ and consider finite and semi-infinite intervals, respectively.

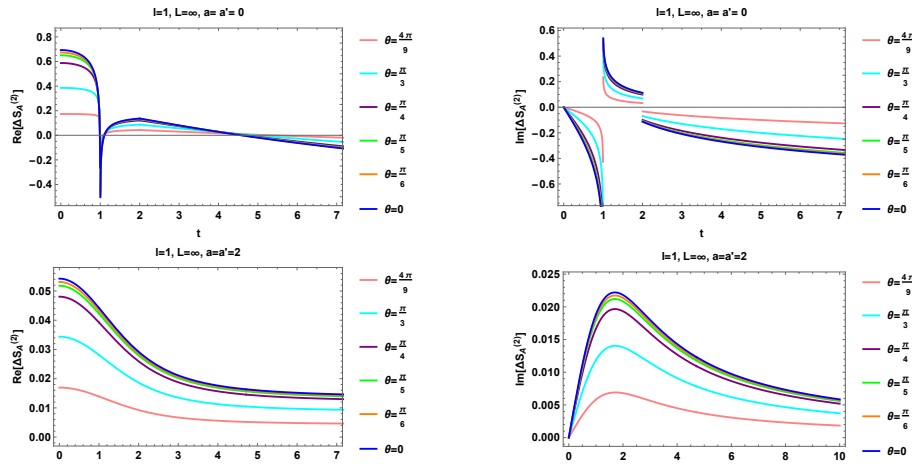

Figure 11: The real and imaginary parts of $\Delta S_A^{(2)}$ for the operator $\mathcal{O}_3$ as a function of $t$ for different values of $\theta$: *First Row)* $a = a' = 0$ *Second Row)* $a = a' = 2$. In all of the figures, we set $L = \infty$ and $l = 1$.

## 4.1 Finite Interval

We first consider a finite entangling region $A \in [0, L]$. By applying eq. (2.6), one can simply find the location of the operators on the complex plane $\mathbb{C}$ as follows (See also ref. [20])

$$z_1 = e^{-\frac{2\pi i}{3}} z_3 = e^{\frac{2\pi i}{3}} z_5 = \left(\frac{w_1}{w_1 - L}\right)^{\frac{1}{3}} = \left(\frac{ia' + t - l}{ia' + t - l - L}\right)^{\frac{1}{3}},$$

$$z_2 = e^{-\frac{2\pi i}{3}} z_4 = e^{\frac{2\pi i}{3}} z_6 = \left(\frac{w_2}{w_2 - L}\right)^{\frac{1}{3}} = \left(\frac{ia + l}{ia + l + L}\right)^{\frac{1}{3}}. \tag{4.1}$$

### 4.1.1 $\mathcal{O}_1$

For the operator $\mathcal{O}_1$, by applying eqs. (2.10) and (2.12), one can simply calculate the two-point function on $\Sigma_1$ as follows

$$\langle \mathcal{O}_1^\dagger(w_1, \bar{w}_1) \mathcal{O}_1(w_2, \bar{w}_2) \rangle_{\Sigma_1} = \left| \frac{(z_1^3 - 1)(z_2^3 - 1)}{L(z_2^3 - z_1^3)} \right|^{4\Delta_1}. \tag{4.2}$$

On the other hand, from eq. (2.16), it is straightforward to show that the six-point function on the complex plane $\mathbb{C}$ is given by

$$\langle \prod_{l=1}^{3} \mathcal{O}_1^\dagger(z_{2l-1}, \bar{z}_{2l-1}) \mathcal{O}_1(z_{2l}, \bar{z}_{2l}) \rangle_{\Sigma_1} = \langle - + - + - + \rangle_{\Sigma_1}$$

$$= \left| \frac{z_{13} z_{15} z_{24} z_{26} z_{35} z_{46}}{z_{12} z_{14} z_{16} z_{23} z_{25} z_{34} z_{36} z_{45} z_{56}} \right|^{\frac{1}{2}}. \tag{4.3}$$

Next, by applying eq. (4.1), it is straightforward to verify the following identities among $z_{ij}$'s

$$|z_{13}| = |z_{15}| = |z_{35}| = \sqrt{3}\,|z_1|, \qquad |z_{24}| = |z_{26}| = |z_{46}| = \sqrt{3}\,|z_2|,$$

$$|z_{12}| = |z_{34}| = |z_{56}|, \qquad |z_{14}| = |z_{25}| = |z_{36}|, \qquad |z_{23}| = |z_{16}| = |z_{45}|. \tag{4.4}$$

Now, by using eq. (4.4), one can rewrite eq. (4.3) as follows

$$\langle\mathcal{O}_1^\dagger(z_1,\bar{z}_1)\mathcal{O}_1(z_2,\bar{z}_2)\mathcal{O}_1^\dagger(z_3,\bar{z}_3)\mathcal{O}_1(z_4,\bar{z}_4)\mathcal{O}_1^\dagger(z_5,\bar{z}_5)\mathcal{O}_1(z_6,\bar{z}_6)\rangle_{\Sigma_1} = \left|\frac{z_{13}z_{24}}{z_{12}z_{14}z_{23}}\right|^{\frac{3}{2}}. \tag{4.5}$$

On the other hand, by applying the conformal transformation in eqs. (2.10), one can write the six-point function on $\Sigma_3$ in terms of the six-point function on $\mathbb{C}$ as follows

$$\langle\prod_{l=1}^{3}\mathcal{O}_1^\dagger(w_{2l-1},\bar{w}_{2l-1})\mathcal{O}_1(w_{2l},\bar{w}_{2l})\rangle_{\Sigma_3} = \prod_{j=1}^{6}\left|\frac{dw_j}{dz_j}\right|^{-2\Delta_1}\langle\prod_{l=1}^{3}\mathcal{O}_1^\dagger(z_{2l-1},\bar{z}_{2l-1})\mathcal{O}_1(z_{2l},\bar{z}_{2l})\rangle_{\Sigma_1}$$

$$= \left|\frac{(z_1^3-1)(z_2^3-1)}{3Lz_1z_2}\right|^{12\Delta_1}\left|\frac{z_{13}z_{24}}{z_{12}z_{14}z_{23}}\right|^{\frac{3}{2}}. \tag{4.6}$$

Next, from eqs. (4.2) and (4.6), one arrive at

$$\frac{\mathrm{Tr}\,(\tau_A)^3}{\mathrm{Tr}\left(\rho_A^{(0)}\right)^3} = \frac{\langle\mathcal{O}_1^\dagger(w_1,\bar{w}_1)\mathcal{O}_1(w_2,\bar{w}_2)\mathcal{O}_1^\dagger(w_3,\bar{w}_3)\mathcal{O}_1(w_4,\bar{w}_4)\mathcal{O}_1^\dagger(w_5,\bar{w}_5)\mathcal{O}_1(w_6,\bar{w}_6)\rangle_{\Sigma_3}}{\left(\mathcal{O}_1^\dagger(w_1,\bar{w}_1)\mathcal{O}_1(w_2,\bar{w}_2)\rangle_{\Sigma_1}\right)^3}$$

$$= \left|\frac{(z_2^3-z_1^3)}{3z_1z_2}\right|^{\frac{3}{2}}\left|\frac{z_{13}z_{24}}{z_{12}z_{14}z_{23}}\right|^{\frac{3}{2}}$$

$$= 1, \tag{4.7}$$

where in the last line we applied eqs. (4.1) and (4.4) (See also eq. (4.12)). Then, from eq. (3.4), one concludes that $\Delta S_A^{(3)} = 0$. Therefore, in the following, we restrict ourselves to operators $\mathcal{O}_{2,3}$.

### 4.1.2 $\mathcal{O}_2$

For the operator $\mathcal{O}_2$, by applying eqs. (2.10) and (2.12), one can simply find the two-point function on $\Sigma_1$ as follows

$$\langle\mathcal{O}_2^\dagger(w_1,\bar{w}_1)\mathcal{O}_2(w_2,\bar{w}_2)\rangle_{\Sigma_1} = \left|\frac{(z_1^3-1)(z_2^3-1)}{L(z_2^3-z_1^3)}\right|^{4\Delta_2}. \tag{4.8}$$

On the other hand, from eq. (2.16), one can obtain the six-point function on the complex plane $\mathbb{C}$ as follows

$$\langle\mathcal{O}_2^\dagger(z_1,\bar{z}_1)\mathcal{O}_2(z_2,\bar{z}_2)\mathcal{O}_2^\dagger(z_3,\bar{z}_3)\mathcal{O}_2(z_4,\bar{z}_4)\mathcal{O}_2^\dagger(z_5,\bar{z}_5)\mathcal{O}_2(z_6,\bar{z}_6)\rangle_{\Sigma_1} = \frac{1}{4}\sum_{j=1}^{10}I_j, \tag{4.9}$$

where $I_j$'s are defined as follows

$$I_1 = \langle---+++\rangle_{\Sigma_1} = \langle+++---\rangle_{\Sigma_1} = \left|\frac{z_{12}z_{13}z_{23}z_{45}z_{46}z_{56}}{z_{14}z_{15}z_{16}z_{24}z_{25}z_{26}z_{34}z_{35}z_{36}}\right|^{\frac{1}{2}},$$

$$I_2 = \langle--+-++\rangle_{\Sigma_1} = \langle++-+--\rangle_{\Sigma_1} = \left|\frac{z_{12}z_{14}z_{24}z_{35}z_{36}z_{56}}{z_{13}z_{15}z_{16}z_{23}z_{25}z_{26}z_{34}z_{45}z_{46}}\right|^{\frac{1}{2}},$$

$$I_3 = \langle--+++-+\rangle_{\Sigma_1} = \langle++---+-\rangle_{\Sigma_1} = \left|\frac{z_{12}z_{15}z_{25}z_{34}z_{36}z_{46}}{z_{13}z_{14}z_{16}z_{23}z_{24}z_{26}z_{35}z_{45}z_{56}}\right|^{\frac{1}{2}},$$

$$I_4 = \langle --++ +- \rangle_{\Sigma_1} = \langle ++--- + \rangle_{\Sigma_1} = \left| \frac{z_{12}z_{16}z_{26}z_{34}z_{35}z_{45}}{z_{13}z_{14}z_{15}z_{23}z_{24}z_{25}z_{36}z_{46}z_{56}} \right|^{\frac{1}{2}},$$

$$I_5 = \langle -+--++ \rangle_{\Sigma_1} = \langle +-++ -- \rangle_{\Sigma_1} = \left| \frac{z_{13}z_{14}z_{25}z_{26}z_{34}z_{56}}{z_{12}z_{15}z_{16}z_{23}z_{24}z_{35}z_{36}z_{45}z_{46}} \right|^{\frac{1}{2}},$$

$$I_6 = \langle -+-+-+ \rangle_{\Sigma_1} = \langle +-+-+- \rangle_{\Sigma_1} = \left| \frac{z_{13}z_{15}z_{24}z_{26}z_{35}z_{46}}{z_{12}z_{14}z_{16}z_{23}z_{25}z_{34}z_{36}z_{45}z_{56}} \right|^{\frac{1}{2}},$$

$$I_7 = \langle -+-++- \rangle_{\Sigma_1} = \langle +-+--+ \rangle_{\Sigma_1} = \left| \frac{z_{13}z_{16}z_{24}z_{25}z_{36}z_{45}}{z_{12}z_{14}z_{15}z_{23}z_{26}z_{34}z_{35}z_{46}z_{56}} \right|^{\frac{1}{2}},$$

$$I_8 = \langle -++--+ \rangle_{\Sigma_1} = \langle +---++- \rangle_{\Sigma_1} = \left| \frac{z_{14}z_{15}z_{23}z_{26}z_{36}z_{45}}{z_{12}z_{13}z_{16}z_{24}z_{25}z_{34}z_{35}z_{46}z_{56}} \right|^{\frac{1}{2}},$$

$$I_9 = \langle -++-+- \rangle_{\Sigma_1} = \langle +---+- \rangle_{\Sigma_1} = \left| \frac{z_{14}z_{16}z_{23}z_{25}z_{35}z_{46}}{z_{12}z_{13}z_{15}z_{24}z_{26}z_{34}z_{36}z_{45}z_{56}} \right|^{\frac{1}{2}},$$

$$I_{10} = \langle -+++ -- \rangle_{\Sigma_1} = \langle +----++ \rangle_{\Sigma_1} = \left| \frac{z_{15}z_{16}z_{23}z_{24}z_{34}z_{56}}{z_{12}z_{13}z_{14}z_{25}z_{26}z_{35}z_{36}z_{45}z_{46}} \right|^{\frac{1}{2}}. \tag{4.10}$$

Moreover, by applying eq. (4.4), one can rewrite eq. (4.10) as follows

$$I_1 = I_4 = I_{10} = \left| \frac{z_{12}z_{23}}{z_{14}^3 z_{13}z_{24}} \right|^{\frac{1}{2}} = \frac{1}{\alpha} \left| \eta_{32}^{56} \right|^2,$$

$$I_2 = I_3 = I_5 = \left| \frac{z_{12}z_{14}}{z_{23}^3 z_{13}z_{24}} \right|^{\frac{1}{2}} = \frac{1}{\alpha} \left| \eta_{56}^{14} \right|^2,$$

$$I_6 = \left| \frac{z_{13}z_{24}}{z_{12}z_{23}z_{14}} \right|^{\frac{3}{2}} = \frac{1}{\alpha},$$

$$I_7 = I_8 = I_9 = \left| \frac{z_{14}z_{23}}{z_{12}^3 z_{13}z_{24}} \right|^{\frac{1}{2}} = \frac{1}{\alpha} \left| \eta_{14}^{32} \right|^2, \tag{4.11}$$

where we defined

$$\alpha = \left| \frac{(z_2^3 - z_1^3)}{3z_1 z_2} \right|^{\frac{3}{2}} = \left| \frac{1}{3} z_{12} \left( 1 + \frac{z_1^2 + z_2^2}{z_1 z_2} \right) \right|^{\frac{3}{2}} = \left| z_{12} \eta_{14}^{32} \right|^{\frac{3}{2}} = \left| z_{12} \frac{z_{14}z_{23}}{z_{13}z_{24}} \right|^{\frac{3}{2}}. \tag{4.12}$$

Moreover, we defined

$$\eta_{ij}^{kl} = \frac{z_{ij}z_{kl}}{z_{ik}z_{jl}}, \tag{4.13}$$

similar to ref. [20]. Then, it is straightforward to show that

$$\eta_{14}^{32} = \frac{z_{14}z_{23}}{z_{13}z_{24}} = \frac{1}{3} \left( 1 + \frac{z_1^2 + z_2^2}{z_1 z_2} \right),$$

$$\eta_{56}^{14} = \frac{z_{56}z_{14}}{z_{15}z_{46}} = \frac{e^{\frac{-\pi i}{3}}}{3z_1 z_2} \left[ -z_1^2 + e^{\frac{\pi i}{3}} z_1 z_2 - e^{\frac{2\pi i}{3}} z_2^2 \right],$$

$$\eta_{32}^{56} = -\frac{z_{23}z_{56}}{z_{35}z_{26}} = \frac{e^{\frac{\pi i}{3}}}{3z_1 z_2} \left[ -z_1^2 + e^{\frac{-\pi i}{3}} z_1 z_2 - e^{\frac{-2\pi i}{3}} z_2^2 \right]. \tag{4.14}$$

On the other hand, by applying the conformal transformation in eqs. (2.10), one can write the six-point function on $\Sigma_3$ in terms of the six-point function on $\mathbb{C}$ as follows

$$\langle\prod_{l=1}^{3}\mathcal{O}_2^{\dagger}(w_{2l-1},\bar{w}_{2l-1})\mathcal{O}_2(w_{2l},\bar{w}_{2l})\rangle_{\Sigma_3} = \frac{1}{4}\left|\frac{(z_1^3-1)(z_2^3-1)}{3Lz_1z_2}\right|^{12\Delta_2}\sum_{j=1}^{10}I_j. \tag{4.15}$$

Next, by utilizing eqs. (4.8), (4.11) and (4.15), one arrives at

$$\frac{\mathrm{Tr}\,(\tau_A)^3}{\mathrm{Tr}\left(\rho_A^{(0)}\right)^3} = \frac{\langle\mathcal{O}_2^{\dagger}(w_1,\bar{w}_1)\mathcal{O}_2(w_2,\bar{w}_2)\mathcal{O}_2^{\dagger}(w_3,\bar{w}_3)\mathcal{O}_2(w_4,\bar{w}_4)\mathcal{O}_2^{\dagger}(w_5,\bar{w}_5)\mathcal{O}_2(w_6,\bar{w}_6)\rangle_{\Sigma_3}}{\left(\mathcal{O}_2^{\dagger}(w_1,\bar{w}_1)\mathcal{O}_2(w_2,\bar{w}_2)\rangle_{\Sigma_1}\right)^3}$$

$$= \frac{1}{4}\left|\frac{(z_2^3-z_1^3)}{3z_1z_2}\right|^{12\Delta_2}\sum_{j=1}^{10}I_j$$

$$= \frac{\alpha}{4}\sum_{i=1}^{10}I_j = \frac{1}{4}\left[1+3\left(\left|\eta_{32}^{56}\right|^2+\left|\eta_{56}^{14}\right|^2+\left|\eta_{14}^{32}\right|^2\right)\right]. \tag{4.16}$$

At the end, from eqs. (3.4), (4.1), (4.14) and (4.16), one obtains

$$\Delta S_A^{(3)} = \frac{1}{2}\log\left[\frac{4}{1+3\left(\left|\eta_{32}^{56}\right|^2+\left|\eta_{56}^{14}\right|^2+\left|\eta_{14}^{32}\right|^2\right)}\right] \tag{4.17}$$

$$= \frac{1}{2}\log\left[\frac{4\left(\frac{(a^2+l^2)(l^2-(ia'+t)^2)}{(a^2+(l+L)^2)((l+L)^2-(ia'+t)^2)}\right)^{\frac{1}{3}}}{\left(\frac{a^2+l^2}{a^2+(l+L)^2}\right)^{\frac{2}{3}}+2\left(\frac{(a^2+l^2)(l^2-(ia'+t)^2)}{(a^2+(l+L)^2)((l+L)^2-(ia'+t)^2)}\right)^{\frac{1}{3}}+\left(\frac{l^2-(ia'+t)^2}{(l+L)^2-(ia'+t)^2}\right)^{\frac{2}{3}}}\right].$$

Moreover, at late times one has

$$\Delta S_A^{(3)} = \frac{1}{2}\log\left[\frac{4}{2+\left(\frac{a^2+l^2}{a^2+(l+L)^2}\right)^{\frac{1}{3}}+\left(\frac{a^2+(l+L)^2}{a^2+l^2}\right)^{\frac{1}{3}}}\right]. \tag{4.18}$$

In Figure 12, we plotted the real and imaginary parts of $\Delta S_A^{(3)}$ as a function of $t$ for different values of the cutoffs $a$ and $a'$. It is observed that the behavior of $\Delta S_A^{(3)}$ is the same as that of $\Delta S_A^{(2)}$ in Figure 3. Now, we want to compare the time dependence of the third PREE with that of the third REE of the states $|\psi\rangle$ and $|\phi\rangle$. For the state $|\psi\rangle$, the density matrix is again given by eq. (3.13). By applying eqs. (2.3) and (2.10), one can simply obtain the insertion points as follows

$$z_1 = e^{-\frac{2\pi i}{3}}z_3 = e^{\frac{2\pi i}{3}}z_5 = \left(\frac{w_1}{w_1-L}\right)^{\frac{1}{3}} = \left(\frac{ia-l}{ia-l-L}\right)^{\frac{1}{3}},$$

$$z_2 = e^{-\frac{2\pi i}{3}}z_4 = e^{\frac{2\pi i}{3}}z_6 = \left(\frac{w_2}{w_2-L}\right)^{\frac{1}{3}} = \left(\frac{ia+l}{ia+l+L}\right)^{\frac{1}{3}}. \tag{4.19}$$

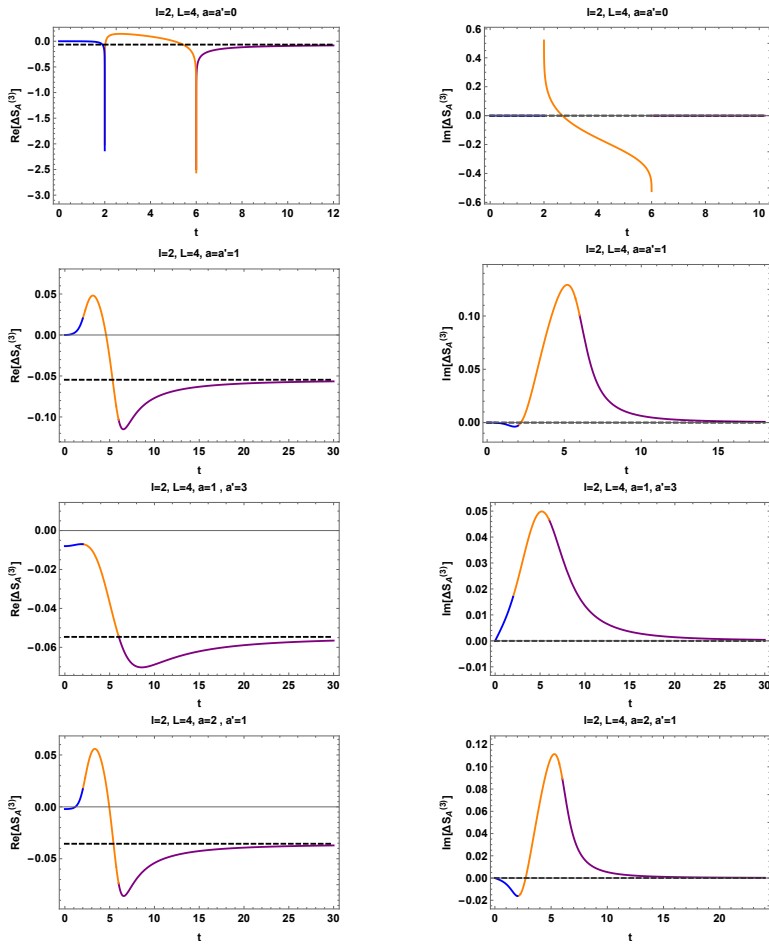

Figure 12: $\Delta S_A^{(3)}$ for the operator $\mathcal{O}_2$, as a function of $t$ for $l = 2$, $L = 4$ and *First Row)* $a = a' = 0$ *Second Row)* $a = a' = 1$ *Third Row)* $a = 1, a' = 3$ *Fourth Row)* $a = 2, a' = 1$. The values of $\Delta S_A^{(2)}$ in the time intervals $0 < t < l$, $l < t < l + L$ and $t > l + L$ are shown in blue, orange and purple, respectively. At late times, it saturates to a constant real value given by eq. (4.18) which is shown by the dashed black line.

It is straightforward to verify that the third REE is always zero for the state $|\psi\rangle$. [12] On the other hand, for the state $|\phi\rangle$, the density matrix is given by eq. (3.13), and hence one arrives at

$$z_1 = e^{-\frac{2\pi i}{3}} z_3 = e^{\frac{2\pi i}{3}} z_5 = \left( \frac{w_1}{w_1 - L} \right)^{\frac{1}{3}} = \left( \frac{ia' + t - l}{ia' + t - l - L} \right)^{\frac{1}{3}},$$

$$z_2 = e^{-\frac{2\pi i}{3}} z_4 = e^{\frac{2\pi i}{3}} z_6 = \left( \frac{w_2}{w_2 - L} \right)^{\frac{1}{3}} = \left( \frac{ia' - t + l}{ia' - t + l + L} \right)^{\frac{1}{3}}. \tag{4.20}$$

In Figure 13, we plotted the third REE $\Delta S_A^{(3)}$ for $|\phi\rangle$ and its time derivative as a function of $t$. It is observed that $\Delta S_A^{(3)}$ is always real and depends on $t$. Moreover, for $a' = 0$, the curve is not continuous. However, for a finite cutoff, the curve is a smooth function of $t$. Next, comparison of Figures 12 and 13, shows that the time dependence of the real part of the third PREE is different from the third REE of the state $|\phi\rangle$. Moreover, the behavior of the imaginary part of the third PREE is not similar to that of the time derivative of the third REE of the state $|\phi\rangle$.

---

[12]The state $|\phi\rangle$ at $t = 0$, is the same as $|\psi\rangle$. Therefore, $\Delta S_A^{(3)}$ for $|\psi\rangle$ has to be equal to that of $|\phi\rangle$ at $t = 0$, which is zero.

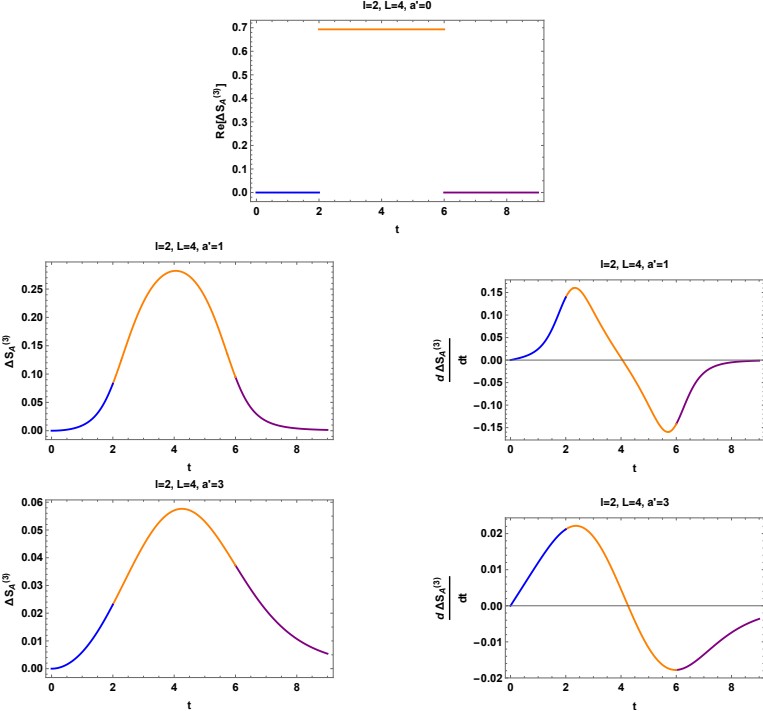

Figure 13: The third REE of the state $|\phi\rangle$, i.e. $\Delta S_A^{(3)}$, as a function of $t$ for $l = 2$, $L = 4$ and *Left)* $a' = 0$ *Middle)* $a' = 1$ *Right)* $a' = 3$. The values of $\Delta S_A^{(3)}$ in the time intervals $0 < t < l$, $l < t < l + L$ and $t > l + L$ are shown in blue, orange and purple, respectively.

In Figures 14, 15 and 16, we plotted $\Delta S_A^{(3)}$ as a function of $x_m$. In Figure 14, we have $a < a'$. This Figure is the same as Figure 6 for $\Delta S_A^{(2)}$. In Figure 15, we have $a > a'$. This Figure is the same as Figure 7. In Figure 16, we have $a = a'$. This Figure is the same as Figure 8. Therefore, from Figures 14, 15 and 16, one can see that $\Delta S_A^{(3)}$ behaves the same as $\Delta S_A^{(2)}$ as a function of $x_m$.

### 4.1.3 $\mathcal{O}_3$

For the operator $\mathcal{O}_3$, the two-point function is as follows

$$
\begin{aligned}
\langle \mathcal{O}_3^\dagger(w_1, \bar{w}_1) \mathcal{O}_3(w_2, \bar{w}_2) \rangle_{\Sigma_1} &= \frac{1}{2}\left[\langle -+\rangle + \cos^2\theta \langle +-\rangle\right] \\
&= \frac{(1 + \cos^2\theta)}{2} \frac{1}{|w_{12}|^{\frac{1}{2}}} \\
&= \frac{(1 + \cos^2\theta)}{2} \left|\frac{(z_1^3 - 1)(z_2^3 - 1)}{L(z_2^3 - z_1^3)}\right|^{\frac{1}{2}}.
\end{aligned}
\tag{4.21}
$$

On the other hand, the six-point function on $\Sigma_3$ is given by

$$
\begin{aligned}
\langle \prod_{l=1}^{3} \mathcal{O}_3^\dagger(w_{2l-1}, \bar{w}_{2l-1}) \mathcal{O}_3(w_{2l}, \bar{w}_{2l}) \rangle_{\Sigma_3} &= \frac{1}{8}\left|\frac{(z_1^3 - 1)(z_2^3 - 1)}{3Lz_1z_2}\right|^{\frac{3}{2}} \Big[I_6\left(1 + \cos^6\theta\right) \\
&\quad + \cos^2\theta\left(1 + \cos^2\theta\right)\left(I_1 + I_2 + I_3 + I_4 + I_5 + I_7 + I_8 + I_9 + I_{10}\right)\Big],
\end{aligned}
\tag{4.22}
$$

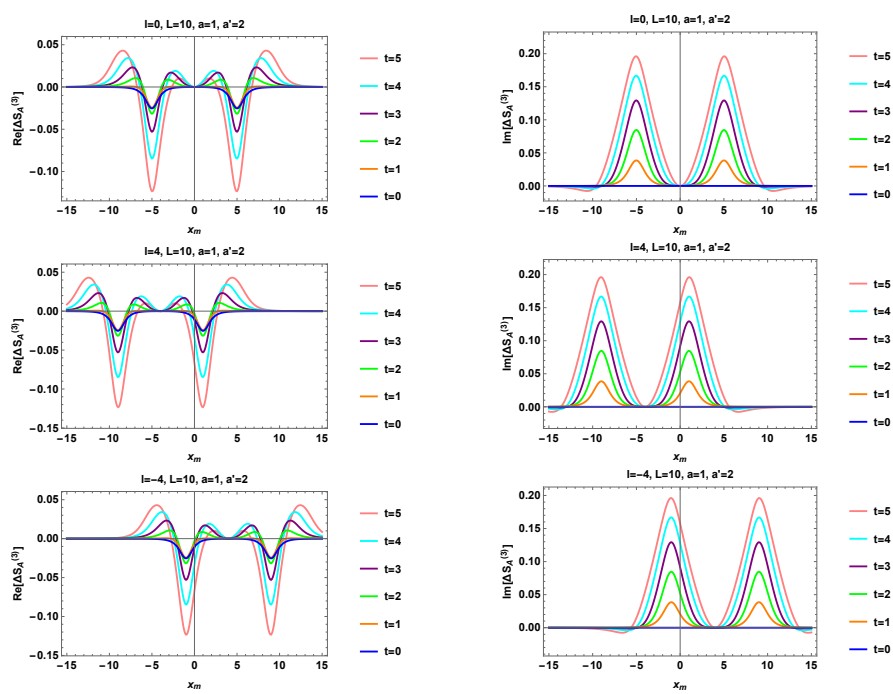

Figure 14: $\Delta S_A^{(3)}$ for the operator $\mathcal{O}_2$, as a function of $x_m$ for $L = 10$, $a = 1$, $a' = 2$ *Top Row)* $l = 0$ *Middle Row)* $l = 4$ *Down Row)* $l = -4$.

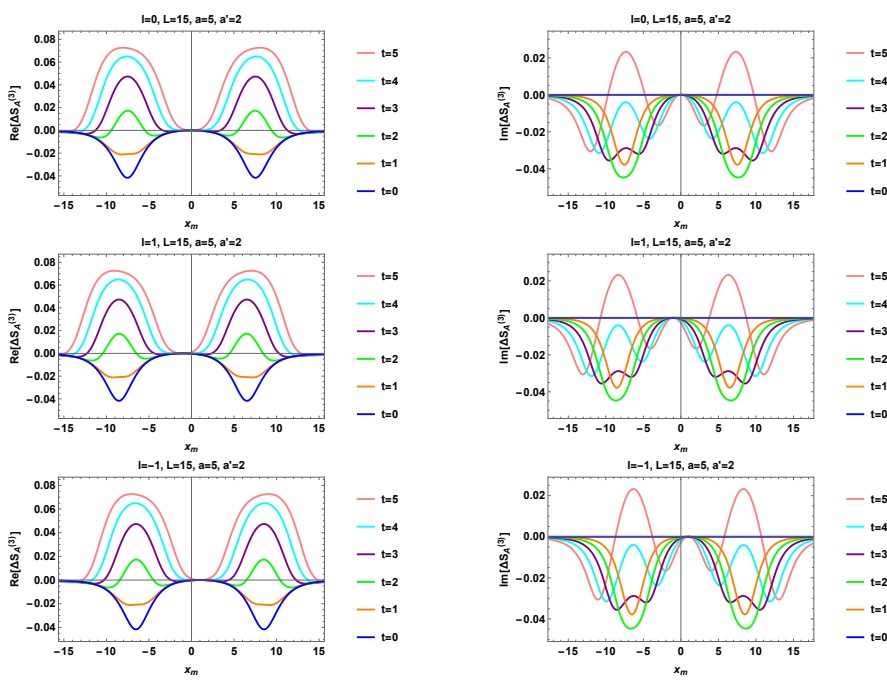

Figure 15: $\Delta S_A^{(3)}$ for the operator $\mathcal{O}_2$, as a function of $x_m$ for $L = 15$, $a = 5$, $a' = 2$ *Top Row)* $l = 0$ *Middle Row)* $l = 1$ *Down Row)* $l = -1$.

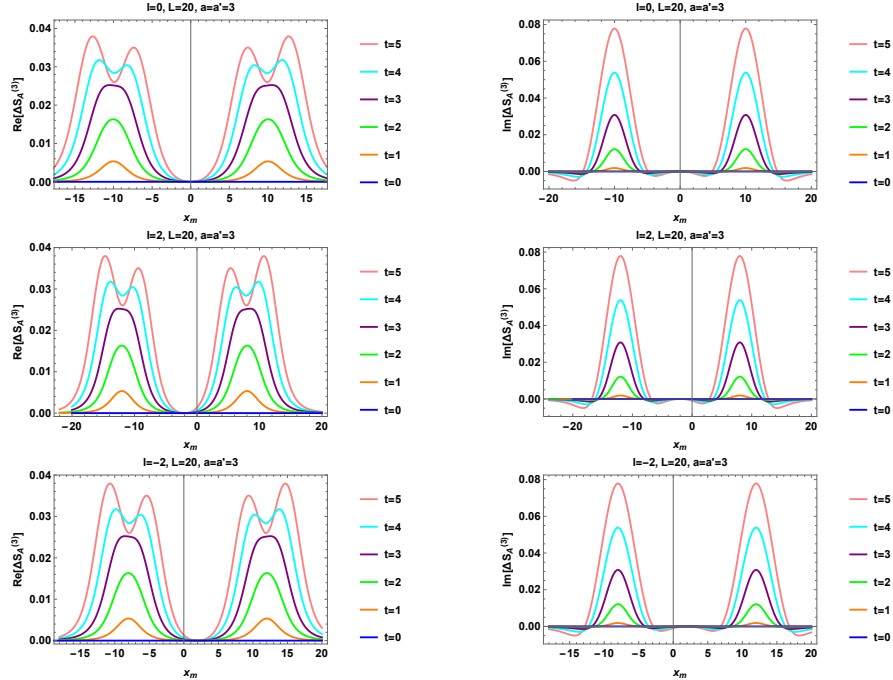

Figure 16: $\Delta S_A^{(3)}$ for the operator $\mathcal{O}_2$, as a function of $x_m$ for $L = 20$, $a = a' = 3$ *Top Row)* $l = 0$ *Middle Row)* $l = 2$ *Down Row)* $l = -2$.

where $I_j$'s are defined in eq. (4.11). Next, by plugging eqs. (4.14), (4.21) and (4.22) into eq. (3.4), one obtains

$$
\begin{aligned}
\Delta S_A^{(3)} &= \frac{1}{2} \log \left[ \frac{(1 + \cos^2 \theta)^3}{1 + \cos^6 \theta + 3 \cos^2 \theta \left( 1 + \cos^2 \theta \right) \left( \left| \eta_{32}^{56} \right|^2 + \left| \eta_{56}^{14} \right|^2 + \left| \eta_{14}^{32} \right|^2 \right)} \right] \\
&= \frac{1}{2} \log \left[ \frac{(1 + \cos^2 \theta)^3}{(1 + \cos^6 \theta) + \cos^2 \theta \left( 1 + \cos^2 \theta \right) C} \right],
\end{aligned}
\tag{4.23}
$$

where we defined

$$
C = \frac{\left( \frac{a^2 + l^2}{a^2 + (l+L)^2} \right)^{\frac{2}{3}} + \left( \frac{(a^2 + l^2)(l^2 - (ia'+t)^2)}{(a^2 + (l+L)^2)((l+L)^2 - (ia'+t)^2)} \right)^{\frac{1}{3}} + \left( \frac{l^2 - (ia'+t)^2}{(l+L)^2 - (ia'+t)^2} \right)^{\frac{2}{3}}}{\left( \frac{(a^2 + l^2)(l^2 - (ia'+t)^2)}{(a^2 + (l+L)^2)((l+L)^2 - (ia'+t)^2)} \right)^{\frac{1}{3}}}
\tag{4.24}
$$

In Figure 17, we plotted $\Delta S_A^{(3)}$ as a function of $\theta$. This Figure is the same as Figure 9 for $\Delta S_A^{(2)}$ of the operator $\mathcal{O}_3$.

## 4.2   Semi-Infinite Interval

In this section, we calculate $\Delta S_A^{(3)}$ for the case where $A \in [0, \infty]$. In this case, from eqs. (2.6) and (2.41), one finds the insertion points of the operators as follows

$$
z_1 = e^{-\frac{2\pi i}{3}} z_3 = e^{\frac{2\pi i}{3}} z_5 = \left( ia' + t - l \right)^{\frac{1}{3}},
$$

$$
z_2 = e^{-\frac{2\pi i}{3}} z_4 = e^{\frac{2\pi i}{3}} z_6 = e^{\frac{\pi i}{3}} \left( ia + l \right)^{\frac{1}{3}}.
\tag{4.25}
$$

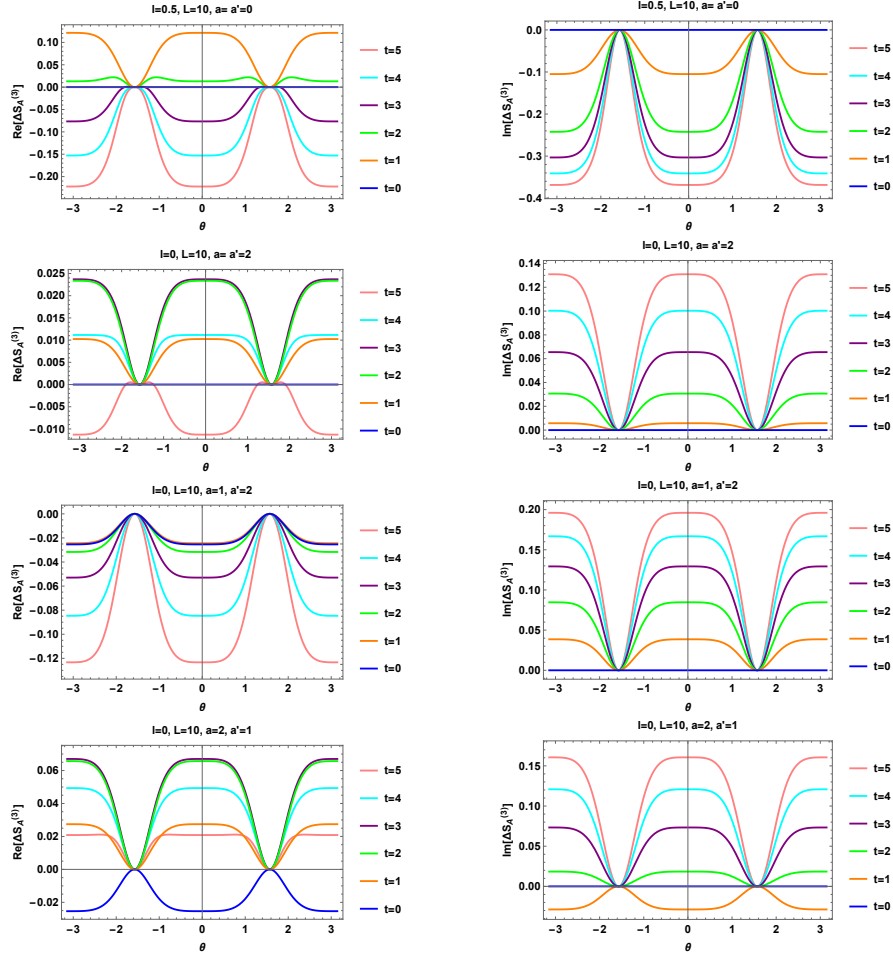

Figure 17: The real and imaginary parts of $\Delta S_A^{(3)}$ for the operator $\mathcal{O}_3$ as a function of $\theta$ for: *First Row)* $a = a' = 0$, $l = 0.5$ *Second Row)* $a = a' = 2$, $l = 0$ *Third Row)* $a = 1$, $a' = 2$, $l = 0$ *Fourth Row)* $a = 2$, $a' = 1$, $l = 0$. In all of the figures, we set $L = 10$.

### 4.2.1   $\mathcal{O}_1$

It is straightforward to check that eq. (4.7) is still valid. Therefore, the third PREE for the operator $\mathcal{O}_1$ is zero again.

### 4.2.2   $\mathcal{O}_2$

For the operator $\mathcal{O}_2$, by plugging eq. (4.25) into eq. (4.17), one has

$$\Delta S_A^{(3)} = \frac{1}{2} \log \left[ \frac{8(a^2 + l^2)^{\frac{1}{3}} \left((ia' + t)^2 - l^2\right)^{\frac{1}{3}}}{(1 - i\sqrt{3})(a^2 + l^2)^{\frac{2}{3}} + (1 + i\sqrt{3})((ia' + t)^2 - l^2)^{\frac{2}{3}} + 4(a^2 + l^2)^{\frac{1}{3}} \left((ia' + t)^2 - l^2\right)^{\frac{1}{3}}} \right].$$

(4.26)

In Figure 18, we plotted $\Delta S_A^{(3)}$ as a function of $t$. This figure is the same as Figure 10 for $\Delta S_A^{(2)}$ of the operator $\mathcal{O}_2$.

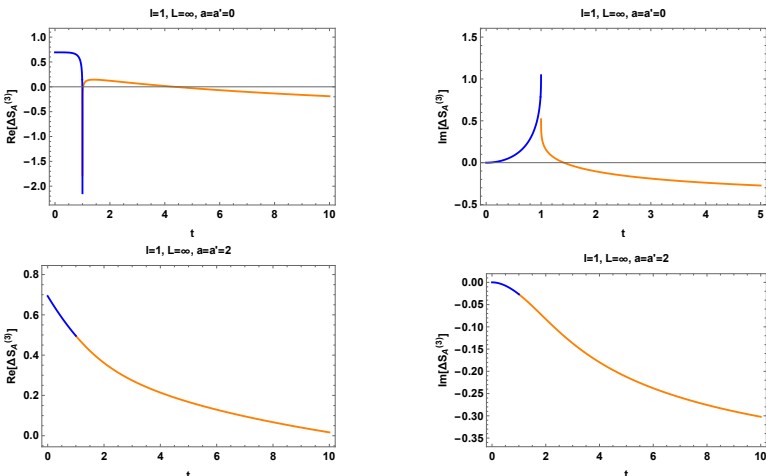

Figure 18: $\Delta S_A^{(3)}$ as a function of $t$ for the operator $\mathcal{O}_2$, $L = \infty$ and *First Row*) $l = 2$ and $a = a' = 0$ *Second Row*) $l = 1$ and $a = a' = 2$. The values of $\Delta S_A^{(3)}$ in the time intervals $0 < t < l$ and $t > l$ are shown in blue and orange, respectively.

### 4.2.3   $\mathcal{O}_3$

For the operator $\mathcal{O}_3$, by plugging eq. (4.25) into the first line of eq. (4.23), one has

$$\Delta S_A^{(3)} = \frac{1}{2} \log \left[ \frac{2(1 + \cos^2 \theta)^2}{2(1 + \cos^4 \theta) + \frac{\left((a^2+l^2)^{\frac{2}{3}}(1-i\sqrt{3})+((ia'+t)^2-l^2)(1+i\sqrt{3})\right)\cos^2 \theta}{(a^2+l^2)^{\frac{1}{3}}((ia'+t)^2-l^2)^{\frac{1}{3}}}} \right]. \tag{4.27}$$

In Figure 19, we plotted $\Delta S_A^{(3)}$ for the operator $\mathcal{O}_3$, as a function of $t$ for different values of $\theta$. Again, we restricted ourselves to the case $0 < \theta < \frac{\pi}{2}$. It is observed that for zero cutoffs, the curves are not smooth. Moreover, the real and imaginary parts of $\Delta S_A^{(3)}$ are neither increasing nor decreasing functions of $\theta$. On the other hand, for non-zero cutoffs, the curves are smooth. Furthermore, the real part of $\Delta S_A^{(3)}$ is neither an increasing nor a decreasing function of $\theta$. However, the imaginary part of $\Delta S_A^{(3)}$ is an increasing function of $\theta$.

## 5   Discussion

In this paper, we investigated the second and third pseudo Rényi entanglement entropies (PREE) $\Delta S_A^{(2,3)}$ for two excited state $|\psi\rangle$ and $|\phi\rangle$ (See eq. (3.1)), in a two-dimensional CFT which is composed of a free massless scalar field. These states are constructed by applying some primary operators $\mathcal{O}_i$ (See eqs. (2.25), (2.26) and (3.2)) on the vacuum state, i.e. $|\psi\rangle = \mathcal{O}_i|0\rangle$. We also included two UV cutoffs $a$ and $a'$ in the states $|\psi\rangle$ and $|\phi\rangle$, respectively (See eq. (3.1)). When the cutoffs $a$ and $a'$ are equal, $|\phi\rangle$ is the time evolution of $|\psi\rangle$. In this manner, by studying the second and third PREEs, we might obtain some information about the the correlation between the state $|\psi\rangle$ and its time evolution. Furthermore, we considered entangling regions which were finite or semi-infinite intervals.

For the operator $\mathcal{O}_1$, the second and third PREEs, i.e. $\Delta S_A^{(2,3)}$, are zero. For the operators $\mathcal{O}_{2,3}$, it was observed that $\Delta S_A^{(2,3)}$ as a function of time $t$ are always complex valued except for $t = 0$ where they are real. Moreover, in the zero cutoff limit, i.e. $a \to 0$ and $a' \to 0$, they are not a smooth function of $t$ at $t = l$ and $t = l + L$ for the finite interval $A \in [0, L]$ and at $t = l$ for the semi-infinite interval

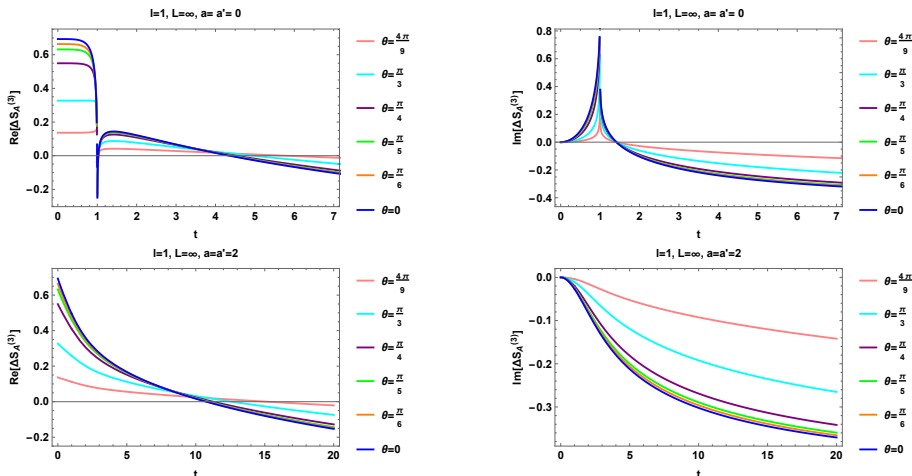

Figure 19: The real and imaginary parts of $\Delta S_A^{(3)}$ for the operator $\mathcal{O}_3$ and the semi-infinite interval $A \in [0, \infty]$ as a function of $t$ for different values of $\theta$: *First Row)* $a = a' = 0$ *Second Row)* $a = a' = 2$. In all of the figures, we set $L = \infty$ and $l = 1$.

$A \in [0, \infty]$ (See the first rows in Figures 3, 10, 12 and 18 for the operator $\mathcal{O}_2$ and the first rows of Figures 17 and 19 for the operator $\mathcal{O}_3$). However, for finite values of the cutoffs $a$ and $a'$, the curves are smooth (See for example the second rows in Figures 3, 10, 12 and 18 for the operator $\mathcal{O}_2$ and the second rows of Figures 11 and 19 for the operator $\mathcal{O}_3$). It should be pointed out that the second and third REE of the excited state $|\phi\rangle$ for the operator $\mathcal{O}_2$ show the same behaviors (See the first rows in Figures 4 and 13).

Furthermore, from the calculation of the PREE for a *global* quench in ref. [24], one might expect that the behaviors of the real (and imaginary) parts of $\Delta S_A^{(2,3)}$ might be similar to those of the second and third Rényi entanglement entropies (and their time derivatives) of the excited states $|\psi\rangle$ and $|\phi\rangle$, respectively. However, by comparison of Figures 3 (12) with 4 (13), respectively, one observes that these behaviors do not happen in our case which we have a *local* quench.

On the other hand, we also plotted $\Delta S_A^{(2,3)}$ for the operator $\mathcal{O}_2$ as a function of the location $x_m$ of the center of a finite entangling region (See eq. (3.18)) for different values of $t$. It was observed that for the operator $\mathcal{O}_2$, $\Delta S_A^{(2,3)}$ are symmetric around the insertion point of the operators at $x = -l$ (See Figures 8, 6, 7, 14, 15 and 16). Moreover, for

- $a < a'$: when the endpoints of the interval coincide the insertion point at $x = -l$, the real part of $\Delta S_A^{(2,3)}$ becomes suppressed very much, such that there are two troughs. As time elapses, the depth of the troughs becomes larger in time. On the other hand, when the endpoints of the interval coincide the insertion point, the imaginary part of $\Delta S_A^{(2)}$ increases very much, such that there are two peaks. As time passes, the heights of the peaks increase with time (See Figures 6 and 14).

- $a > a'$: when the endpoints of the interval coincide the insertion point at $x = -l$, both the real and imaginary parts of $\Delta S_A^{(2,3)}$ are suppressed at early times. However, as time elapses they increase such that two peaks appear. Moreover, the heights of the peaks increase in time (See Figures 7 and 15).

- $a = a'$: when the endpoints of the interval coincides the insertion point at $x = -l$, both the real and imaginary parts of $\Delta S_A^{(2,3)}$ peak. The heights of the peaks grows in time (See Figures 8 and

16).

For the operator $\mathcal{O}_3$, we also studied the behaviors of $\Delta S_A^{(2,3)}$ as a function of $\theta$ for different values of $t$ and the cutoffs (See Figures 9, 17 for the finite interval and Figures 11 and 19 for semi-infinite interval cases). It was observed that for $\theta = \{0, \frac{\pi}{2}\}$, they are zero which is due to the fact that for these vales of $\theta$, the states $|\psi\rangle$ and $|\phi\rangle$ are separable.

There are many interesting directions to pursue in the future: In refs. [9,11], the REE of the excited states constructed by the operators $\mathcal{O}_{1,2}$ were studied in four and six dimensions. It would be interesting to generalize these calculations to higher dimensional CFTs. Moreover, in ref. [33] the PREE was calculated for some descendant operators. It would be interesting to calculate PREE for those descendant operators in our setup and compare the results with those for primary operators. Furthermore, the effect of $T\overline{T}$-deformation on the PREE was investigated in ref. [19]. It would be interesting to investigate the effect of $T\overline{T}$-deformation in our setup. Another direction is to study the effect of temperature on PREE. It should be pointed out that the second REE of some excited sates in ref. [13] was studied and observed that it decreases at late times by increasing the temperature. This suppression was interpreted as a result of decoherence [13]. Notice that the transition matrix is defined for two different pure states. As long as we know, there is no such definition at finite temperature and it is still an open question. [13] [14]

# Acknowledgment

We would like to thank Mohsen Alishahiha very much for his illuminating and helpful comments during this work. We are also very grateful to Masahiro Nozaki, Tadashi Takayanagi, Song He, Kento Watanabe, Ali Mollabashi, Jia Tian, Ali Naseh and Amin Faraji Astaneh for their valuable comments. The work of FO is supported by the Institute For Research In Fundamental Sciences and Iran Science Elites Federation (ISEF). We would also like to thank the Kavli Institute For Theoretical Sciences (KITS) at the University of Chinese Academy of Sciences (UCAS) for its financial supports and warm hospitality when this work was at its final stages. We would also like to thank the organizers and participants of "Beijing Osaka String/Gravity/Black Hole Workshop" at KITS where this work was presented, for their very helpful comments.

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
