# Peer review of "Pseudo Rényi Entanglement Entropies For an Excited State and Its Time Evolution in a 2D CFT"

_SciPost Physics_

## Round 2 · Referee Report · Anonymous (Referee 1) · 2024-3-12

Strengths

The paper titled "Pseudo Rényi Entanglement Entropies For an Excited State and Its Time Evolution in a 2D CFT," authored by Farzad Omidi is a detailed exploration into the second and third pseudo Rényi entanglement entropies (PREE) for a locally excited state and its time evolution in a two-dimensional conformal field theory (CFT2). It focuses on excited states created by applying primary operators on the vacuum state at time zero and investigates their time evolution in terms of PREE for both finite and semi-infinite intervals at zero temperature. The study reveals the complex nature of PREE over time and discusses its dependence on the spatial location of the entangling region's center.

The paper presents significant advancements in understanding the behavior of pseudo Rényi entanglement entropies (PREE) for excited states in two-dimensional conformal field theory. It meticulously examines the second and third PREE for locally excited states, providing insights into their evolution over time. The findings highlight a complex dependence of PREE on the temporal and spatial parameters, offering new perspectives on the entanglement dynamics in quantum field theories. This work is pivotal for theoretical research in quantum information and conformal field theory, marking a step forward in entanglement entropy studies.

Weaknesses

  1. The authors can comment on the possible physical meaning or interpretation of the complex value of the PREE.
  2. The figures show the Z_2 symmetry, e.g. figure 9. There are some hidden symmetry for the initial setup and the authors should comment on this issue.

Report

After resolving the issues mentioned in weaknesses, we can make a final decision.

Requested changes

improve the grammar.

---

## Round 2 · Referee Report · Anonymous (Referee 2) · 2024-3-17

Report

This manuscript makes significant progress in the explicit calculation of Renyi Pseudo entropy for several states in the free scalar 2d CFT. The calculations themselves are non-trivial and interesting. The author has made significant contributions with exceptionally clear communication .

Unfortunately, despite the originality and technical achievements of these main calculations there is overall very little analysis of the results beyond the presented figures. Specifically, there is missing a more global discussion as to how these results improve our understanding of PEE and contribute to furthering this field.

Put bluntly for someone working on or interested in PEE it is not clear currently what they would learn or take away from the presented examples. As written this work does not seem to provide any significant improvement in the understanding of the properties of PEE or inspire new research directions.

I believe these issues are fully rectifiable and once done that this paper will be of substantial utility to the scientific community. Provided these concerns can be satisfactorily addressed I would then recommend it for publication.

Requested changes

  1. Significant expansion of the text regarding figures 3-19 as well as discussion section:

a. Justification for why these specific examples were chosen and why they are interesting to consider (This needs to be broader than "because they could be calculated"). b. Given the results (specifically each figure) what does one learn about PEE? What global properties or lessons can one learn. How can these be used to inform future research on PEE?

  1. This in minor but the word "a" in the title is ambiguous. This should be changed to specifically mention the free massless scalar CFT.

---

## Round 2 · Referee Report · Anonymous (Referee 3) · 2024-3-21

Strengths

1- Pedagogical presentation of the technicalities.

Weaknesses

1- Lack of clear interpretation of the results and lack of clear take-home messages.

Report

The manuscript addresses specific calculations of pseudo-Renyi (n=2,3) entropies for specific states in 2d bosonic free CFTs. More precisely, the manuscript is a pseudo-entropy generalization of a series of works originally introduced in [9-11], investigating the time evolution of Renyi entropies followed by a local operator quench in free 2d CFT.

The calculations are valuable because there are very few setups that such an investigation is possible analytically. On the other hand, although valuable observations have been reported, the analysis lacks conclusive lessons about the presented results.

I believe that these calculations have the potential to become significant in the future when the knowledge about (the time evolution of) pseudo-entropy is improved, while the current version of the manuscript needs a significant improvement in the interpretation of the results.

Requested changes

1- Elaboration on the physical significance of the results. For instance, I suggest addressing possible comments about understanding the results in terms of free streaming quasi-particles.

2- Eq. 2.18 is a direct result of conformal symmetry while Eq. 2.16 is a specific result corresponding to vertex operators. The sentence above Eq. 2.18 stating that one needs 2.16 to arrive at 2.18 does not seem to be correct. Eq. 2.16 plays a role after introducing O_1 and O_2 which are composed of vertex operators in Eq. 2.25-26.

---

## Editorial Decision

awaiting_resubmission